# Hierarchical and nonhierarchical features of the mouse visual cortical network

Rinaldo D. D'Souza[1], Quanxin Wang[1,2], Weiqing Ji[1], Andrew M. Meier[1], Henry Kennedy[3,4], Kenneth Knoblauch [3,5] & Andreas Burkhalter [1✉]

Neocortical computations underlying vision are performed by a distributed network of functionally specialized areas. Mouse visual cortex, a dense interareal network that exhibits hierarchical properties, comprises subnetworks interconnecting distinct processing streams. To determine the layout of the mouse visual hierarchy, we have evaluated the laminar patterns formed by interareal axonal projections originating in each of ten areas. Reciprocally connected pairs of areas exhibit feedforward/feedback relationships consistent with a hierarchical organization. Beta regression analyses, which estimate a continuous hierarchical distance measure, indicate that the network comprises multiple nonhierarchical circuits embedded in a hierarchical organization of overlapping levels. Single-unit recordings in anaesthetized mice show that receptive field sizes are generally consistent with the hierarchy, with the ventral stream exhibiting a stricter hierarchy than the dorsal stream. Together, the results provide an anatomical metric for hierarchical distance, and reveal both hierarchical and nonhierarchical motifs in mouse visual cortex.

[1] Department of Neuroscience, Washington University School of Medicine, St. Louis, MO 63110, USA. [2] Allen Institute for Brain Science, Seattle, WA 98109, USA. [3] Stem Cell and Brain Research Institute, Université Lyon, Université Claude Bernard Lyon 1, INSERM, Bron, France. [4] Institute of Neuroscience, State Key Laboratory of Neuroscience, Chinese Academy of Sciences Key Laboratory of Primate Neurobiology, Shanghai 200031, China. [5] National Centre for Optics, Vision and Eye Care, Faculty of Health and Social Sciences, University of South-Eastern Norway, Hasbergsvei 36, 3616 Kongsberg, Norway.
✉email: burkhala@wustl.edu

Visual perception is accomplished in part by generative intracortical mechanisms that produce invariant object recognition, provide contextual influences, and compare incoming information with prior knowledge[1,2]. The classical notion of a cortical hierarchy posits that signals from primary visual cortex (V1) are routed through increasingly specialized cortical areas by ascending feedforward (FF) pathways, whereas descending feedback (FB) pathways selectively shape cortical responses within the receptive field (RF) depending on past experiences and task demand[1,3–5]. Accordingly, each higher area is thought to assemble the feature-selective RFs of lower areas and integrate them to form increasingly complex representations of the visual scene whose diverse spatiotemporal features are distributed across multiple higher-order areas[3,6]. While this bottom-up view of visual processing has inspired theoretical models of object recognition and categorization[2,7], it fails to account for the densely reciprocally interconnected network of the neocortex as well as the inferential nature of visual perception[8–12]. Furthermore, it is unclear if a sequential ranking of areas, as proposed by several hierarchical models[4,7,13,14], adequately describes the visual cortical network or whether the visual system includes non-hierarchical motifs that involve communication between areal pairs that do not exhibit conventional FF and FB relationships[15].

Because FF and FB pathways project to different cortical layers, the laminar architecture of interareal pathways is thought to be essential for hierarchical processing in primates and rodents[4,16]. Thus, pairwise comparisons of distinct FF and FB signatures allow the construction of sequences of interactions between functionally specialized areas. While interactions between hierarchical levels could conceivably occur in discrete steps, interareal processing can alternatively be graded, relying on connections that are neither completely FF nor FB. This necessitates assigning hierarchical distance values between levels and ranking areas in the context of all possible interareal connections[17–19] rather than by pairwise comparisons using qualitative criteria[4]. In primate visual cortex, this procedure drastically reduces the number of solutions for ordering areas[17,18,20]. Here we show that in mouse, a similar procedure provides a unique insight into the hierarchical network. Specifically, it allows addressing the extent to which pairwise hierarchical interactions between areas A and B, and nonhierarchical interactions between B and C coexist.

A recent study, which analyzed laminar patterns of axonal projections throughout the mouse neocortex, generated a hierarchy of the cortical and thalamocortical network[14]. The study designated a FF or FB label to each laminar pattern such that the consistency of hierarchical relationships was maximized; however, the analysis did not account for lateral connections – i.e., connections between areas occupying the same hierarchical level[14]. Thus, it is unclear to what extent consistent hierarchical relationships govern the 'ultra-dense' mouse cortical graph in which almost all possible connections between visual areas have been shown to exist[8,14,21]. Such a dense network could lead to reciprocally connected pairs that exhibit FF laminar patterns in both directions being more frequent than in macaque[19].

Here, we investigated the degree to which visual areas fit into a consistent sequence of levels by examining anterogradely labeled interareal projection patterns. The results show that reciprocal connections on average adhere to an essential hierarchical rule: the more FF a pathway is in one direction, the more emphasized is the FB nature of the reciprocal pathway. By employing a regression model to determine hierarchical level and distance values, we show that while the network exhibits hierarchical features, it is characterized by numerous lateral connections, and is best described as comprising five overlapping levels. Single-unit recordings of RF sizes in different areas further show that the structural hierarchy is consistent with the physiological hierarchy.

## Results

**Diverse laminar projection patterns across cortico-cortical pathways.** FF and FB axonal projections in the mouse visual cortex exhibit distinct laminar termination patterns[14,22]. To evaluate these patterns, we injected the anterograde tracer BDA into V1 and 9 of the 10 higher visual areas that have been previously identified through topographic mapping of projections from V1[23,24], and whose borders have been identified through anatomical and molecular landmarks:[8,21,25] LM (lateromedial), AL (anterolateral), RL (rostrolateral), P (posterior), LI (laterointermediate), PM (posteromedial), AM (anteromedial), A (anterior), POR (postrhinal), and PORa (anterior POR) (Fig. 1a, b). Each area was injected twice in two different animals ($n = 23$ mice). In order to minimize BDA uptake by broken fibers of passage, and to ensure labeling of neurons in all six layers, iontophoretic injections with fine pipettes were performed through the depth of cortex. Laminar patterns of axonal projections in the nine target areas were examined in the coronal plane. Coronal sections were numbered beginning from the posterior pole of occipital cortex, and projections to individual areas were identified by their locations relative to landmarks formed by bisbenzimide labeled, callosally projecting neurons as well as by their locations relative to each other (Fig. 1a).

Callosal projection landmarks were observed in situ and identified in coronal sections. A band of callosally projecting neurons determined the boundary between V1 and an acallosal zone on the lateral side that included areas LM, LI, and AL (Fig. 1a, inset). This caudo-rostral band can be identified in coronal sections as a vertical column forming the lateral boundary of V1. Within the acallosal zone, LM and AL were respectively identified as distinct posterior and anterior projections. AM was identified as the anterior part of the acallosal region on the medial side of the heavily myelinated V1. PM was contained in the posterior region of this zone.

An example of an injection into area AM is illustrated in Fig. 1a, b. Projections from AM were densest in layer (L) 1 and the middle layers comprising L2/3 and L4 (L2-4) in PM and P, while showing a preference for targeting L1 over L2-4 in V1, LM, AL, and RL (Fig. 1c). AM also densely innervated L5 and L6 of V1 and LM, and targeted all six layers in AL and A. Thus, projections from the same area showed diverse laminar patterns depending on the target area. Representative termination patterns in target areas for an injection in V1 (Fig. 2a, b) and for injections in LM, RL, PM, P, AL, LI, A, and POR show striking laminar differences across pathways (Fig. 2c, d and Supplementary Figs. 1 and 2). These patterns include the dense targeting of all six layers or L1-5 (e.g., RL → AL and P → LM, Supplementary Fig. 1), a preference for targeting L1 over L2-4 (e.g., projections from POR, Supplementary Fig. 2), and the stronger targeting of superficial layers over deep layers (e.g., P → PM, P → A, AL → PM; Supplementary Figs. 1 and 2). We hypothesized that these laminar patterns are constrained by hierarchical distance rules[17–19].

A critical feature of a cortical hierarchy is the presence of identifiable FF and FB pathways. To identify the respective anatomical signatures of FF and FB pathways, we first focused on the termination patterns of projections emanating from, and terminating in, V1. As the primary geniculo-cortical target, V1 can be regarded as the lowest area of the visual cortical hierarchy; accordingly, projections originating in V1 and terminating in the other areas were classified as FF projections, whereas the respective reciprocal connections descending to V1 were regarded as FB. V1 projections to higher areas terminated in a column with relatively sparse terminations in L1 (Fig. 2a, b). In contrast, FB pathways to V1, originating in each of the higher areas, frequently terminated in a distinctly bistratified pattern with preferential

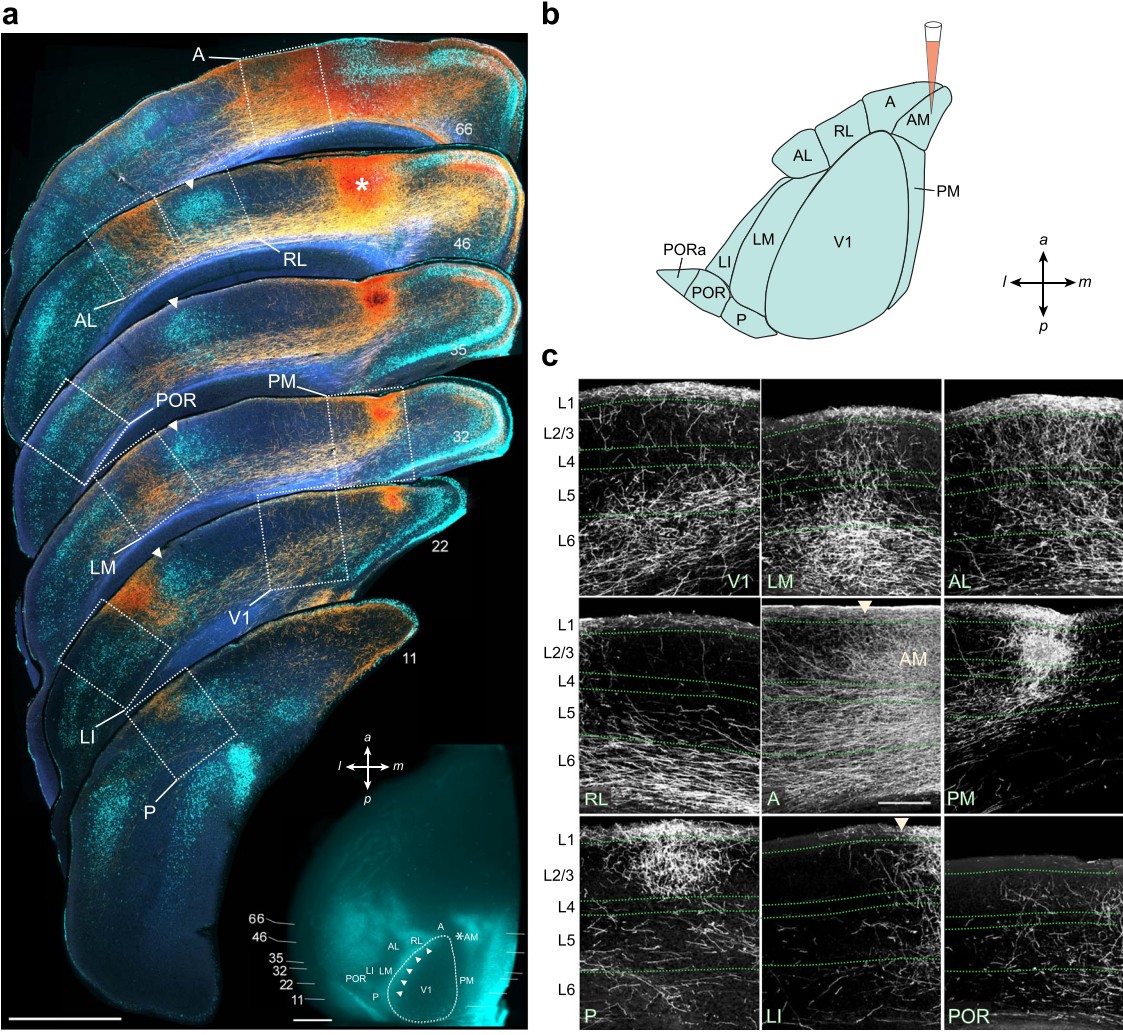

**Fig. 1 Identification of visual areas. a** Rostrocaudal series of coronal sections of the left hemisphere in which AM was injected with BDA. Dark-field images of anterogradely labeled axonal projections (yellow/orange) to distinct visual areas (white outlined boxes). Numbers denote the coronal plane corresponding to the respective rostrocaudal location shown in the inset. Fluorescent images of retrogradely labeled callosal neurons (cyan), after injection of bisbenzimide into the right hemisphere, aid in the identification of areal borders. For example, the column of callosal neurons (arrowheads in coronal sections) corresponds to the band shown in the inset (arrowheads). Inset, In situ image of left hemisphere, before coronal sectioning, showing retrogradely labeled callosally projecting neurons (cyan). Asterisk denotes injection site in AM. White arrowheads indicate a band of callosal neurons that form the boundary between V1 and an acallosal zone that includes LM, LI, and AL. Horizontal lines and numbers denote the coronal planes shown above. Scale bars, 1 mm. **b** Diagram of visual areas and a BDA injection into AM; a anterior, m medial, p posterior, l lateral. **c** High magnification images of regions within white boxes in Fig. 1a. Axonal projections from AM target the other nine areas with varying strengths and laminar patters, and are observed in all six layers. Arrowhead in the LI panel denotes boundary between LI and LM. Arrowhead in the A panel denotes boundary between A and AM. Scale bar, 200 µm.

targeting of L1, L5, and L6, and substantially weaker projections to L2-4 (Fig. 2c).

**Hierarchical features in the mouse visual cortical network.** In order to quantify the polarity or hierarchical distance (i.e. the FF or FB nature) of pathways, we measured the optical density ratio (ODR), defined here as the ratio of the optical density of labeled axons in L2-4 to that in L1 + L2-4, for each connection. This ratio was chosen because (i) the density of projecting axons in L2-4, relative to that in L1, provided a clear distinction between FF and FB pathways, evident in the projections to and from V1 (Fig. 2b, c); and (ii) by excluding deep layers as classifiers, we eliminated the possible contamination by fibers of passage in these layers[26] (Fig. 1a). We reasoned that the ODR would provide a graded hierarchical index that scales across pathways from the most FF to the most FB as has been shown in primate cortex using

retrograde tracers[17–19]. Additionally, ODRs would also allow us to identify strict hierarchical sequences, and determine to what degree the network is hierarchical or nonhierarchical. Out of the 90 possible cortico-cortical connections between any two of the ten injected areas, 80 pathways exhibited projections in the target area that were sufficiently dense to allow analysis.

To calculate the ODR, pixels within the highest 70% intensity values were selected for analyses (Fig. 2a). The ODRs for representative laminar patterns are shown in Fig. 2b–d, which illustrate the relatively high ODRs (0.61–0.88) for FF projections from V1 and low FB ODRs (0.14–0.48). Interareal connections between higher areas typically exhibited columnar termination patterns with variable projection densities in all six layers, and with intermediate ODR values (0.35–0.58; Fig. 2d).

For each pathway, the ODRs from 3 to 5 coronal sections containing the center of the projection to the target area were averaged. After analysis of the laminar patterns in all coronal

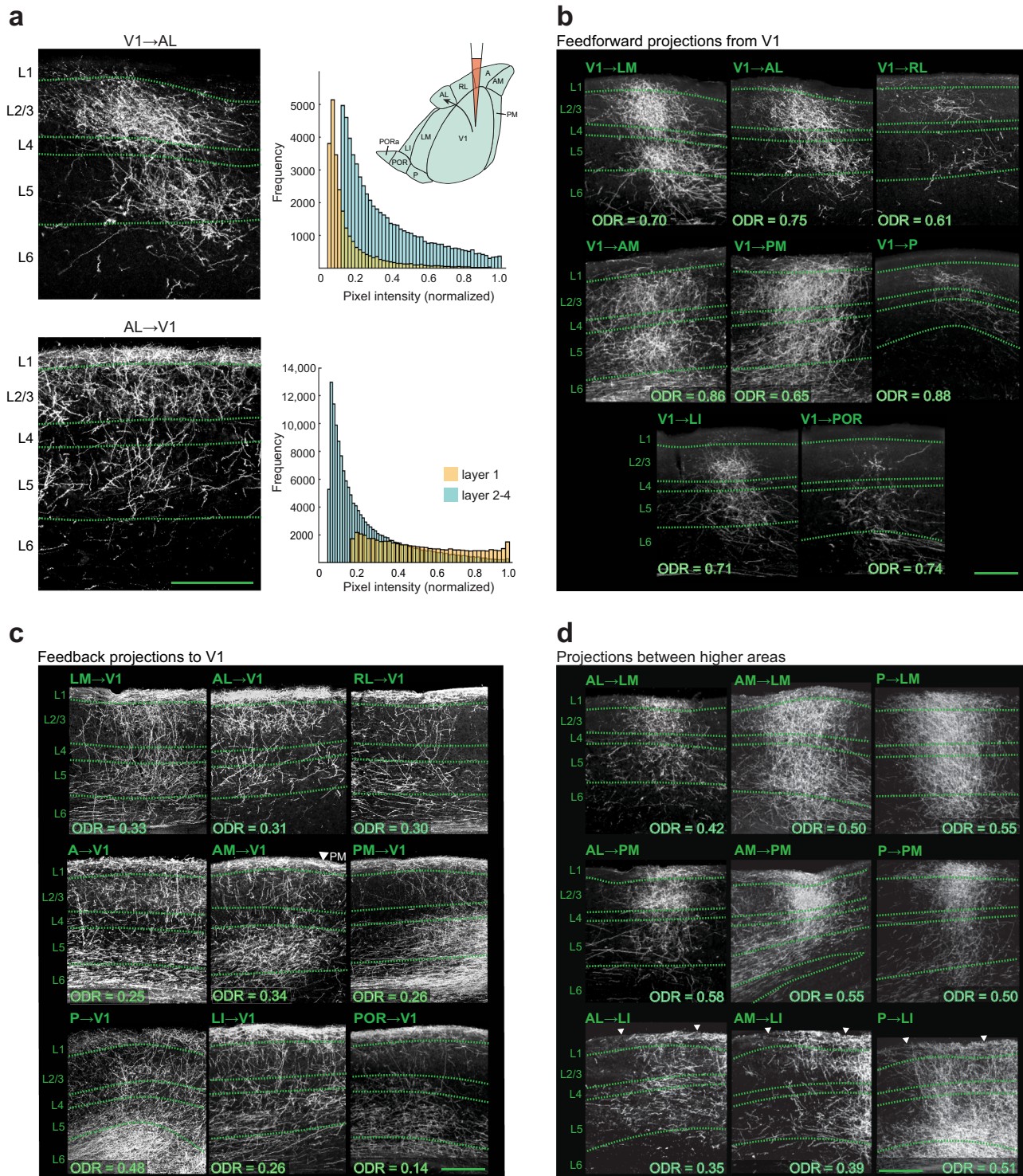

sections, a 10 × 10 matrix of average ODRs for the 80 connections was generated (Fig. 3a). Among the higher visual areas, projections from area LM had, on average, the highest ODR, consistent with the notion that LM may be considered the mouse analog of the primate secondary visual cortex[23]. Terminations of projections from areas LI, AM, and POR, on the other hand, exhibited the lowest mean ODR values indicative of their constituting higher visual areas. The matrix thus indicates the features of a hierarchically organized system in which the relative projection weight of L2-4 and L1 gradually changes.

Because an ODR is a proportion of axonal projections to L2-4 relative to L1-4, all ODRs lie within the standard unit interval (0,1). To obtain a quantitative measure of hierarchical distance between any two areas, a logit transformation of the ODRs was performed so that the ratios were mapped from a (0,1) interval onto the real number line (-∞, ∞) (Supplementary Fig. 3a). This was done so that differences between hierarchical levels remained linear across interareal connections[19]. Figure 3b shows the range and frequency distribution of the logit transformed ODRs of laminar patterns from all coronal sections. Positive and negative

**Fig. 2 Dark-field images of coronal sections showing diverse laminar termination patterns and optical density ratios of intracortical axonal projections.**
**a** Left. Representative termination patterns of the V1 → AL (top) and AL → V1 (bottom) pathways. Right. Histograms showing the distribution of pixel values in L1 (yellow) and L2-4 (blue) in the corresponding dark-field images. Only pixels within 70% of the highest pixel value were included for analyses, and plotted after subtraction of background intensity. Note the overall brighter pixels in L2-4 in the feedforward (FF) V1 → AL pathway ($p < 10^{-16}$, K-S test) and the overall higher pixel values in L1 in the feedback (FB) AL → V1 pathway ($p < 10^{-16}$, K-S test). Top inset. Diagram of an injection into V1 and the anterogradely labeled V1 → AL pathway (arrow). **b**. Laminar termination patterns of FF axonal projections in each higher visual area after injection of BDA into V1. The ratio of the average optical density of axonal projections in L2-4 to that in L1 + L2-4 (optical density ratio, ODR) for each pattern is presented in the respective panel. **c**. Laminar termination patterns of FB projections in V1 after injection of BDA into each of the nine higher visual areas. One injection was performed in each animal. The ODR for each pattern is indicated in the corresponding panel. Arrowhead in the AM → V1 panel demarcates the boundary between V1 and PM. **d**. Nine representative examples of higher visual cortico-cortical laminar termination patterns, for injections of BDA performed in areas P, AL, and AM. The ODR for each pattern is presented in each panel. Arrowheads in the P → LI, AL → LI, and AM → LI panels demarcate the boundaries of LI used for analysis. Scale bars (**a–d**), 200 μm.

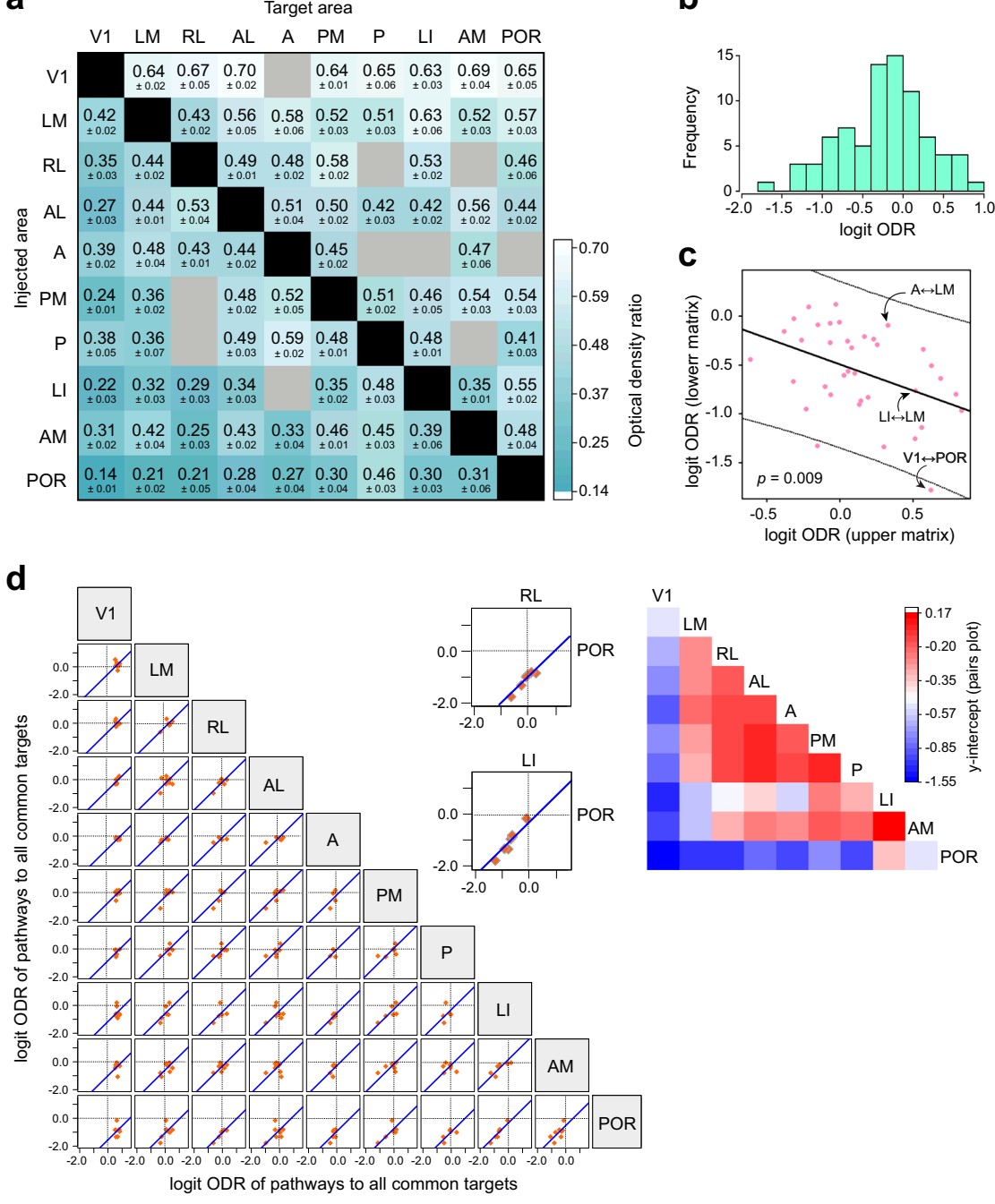

**Fig. 3 Mouse cortical network exhibits hierarchical features. a** 10 × 10 connection matrix of interareal connections between the 10 visual areas. Each block shows the ODR for the respective pathway in which the source and target areas are respectively denoted on the left of each row and the top of each column. Gray blocks represent pathways that could not be analyzed due to weak axonal projections in the target area. **b** Distribution of logit transformed values of the ODRs for all 80 pathways. A logit ODR value of zero indicates an ODR of 0.5. **c** Logit ODR for each pathway plotted against that of its reciprocal counterpart for all 74 pathways that have a dense reciprocal connection (see Fig. 3a). 'Upper matrix' and 'lower matrix' refer respectively to the ODR values in the upper/right and lower/left triangular halves of the matrix in Fig. 3a. The fit shows a significant negative association (slope = −0.53, $p = 0.009$, F-test, F-statistic: 7.54 on 1 and 35 degrees of freedom) indicating that the more FF a pathway is in one direction, the more FB is the reciprocal pathway. The identities of three representative reciprocal connections in the scatterplot are shown to illustrate the variation in asymmetry of ODRs for reciprocally connected areas. **d**. Scatterplots showing the correlation of logit ODR values in all shared targets of any two injected areas. The horizontal axis of each plot corresponds to the logit ODR of the pathway originating in the area indicated at the top of each column, and the vertical axis corresponds to the logit ODR of the pathway originating in the area indicated at the right of each row. Thus, each orange data point plots against each other the logit ODRs for pathways that terminate in a common target area for the corresponding two injected areas. Areas that exhibited weak or absent projections from one of the two injected areas (gray blocks in Fig. 3a) were excluded. Dotted lines, coordinate axes. A line of unit slope (blue) that best fit the points is plotted in each graph, and the absolute value of the y-intercept of this unit line provides a measure for hierarchical distance between the two injected areas. Two example graphs are shown at higher magnification with the injected areas indicated. Note that the absolute value of the y-intercept in the graph plotting pathways originating in RL and POR (y-intercept, −1.04) is greater than that in the graph for pathways from LI and POR (y-intercept, −0.33). This indicates that RL and POR are more hierarchically distant than are LI and POR when only projections emerging from these three areas are considered. The y-intercepts of each graph are plotted as a heat map on the right.

logit ODR values respectively correspond to ODRs larger and smaller than 0.5 (i.e., a probability of 0.5 corresponds to a logit of 0). Thus, the majority of pathways exhibited stronger axonal terminations in L1 than in L2-4, pointing to L1 as a prominent target for cortico-cortical connections.

Importantly, a hierarchy would be expected to exhibit a systematic and consistent gradient of FF and FB relations between any two of the areas forming the network. Specifically, if a pathway is FF in one direction, the reciprocal pathway is expected to be FB, and likewise the hierarchical distance between pairs of areas is expected to be consistent with the hierarchical positioning of each area in the network. For all 37 pairs showing clearly defined reciprocal connections, we plotted the logit transformation of the ODR in one direction against that in its reciprocal direction (Fig. 3c). This shows a negative association ($p = 0.009$, F-test; $n = 74$ pathways) indicating that the more FF a pathway is in one direction, the more FB the reciprocal connection is on average in the opposite direction, implying an ordered arrangement of areas. A similar negative association was observed for the raw ODRs without the logit transformation ($p = 0.009$, Supplementary Fig. 3b). Despite these hierarchical features, the broad spread of the scatterplot points to the existence of a large number of reciprocal connections that do not adhere to strict FF-FB relationships.

Next, we examined whether hierarchical distance values between any two areas reflected their respective positions within the hierarchy. We reasoned that interareal hierarchical distances should be independent of injection site location and therefore consistent across injections; in other words, in a strict hierarchy, the hierarchical distance between two injected areas $i$ and $j$ should be independent of whether we assess laminar patterns terminating in a common target area $p$ or another common target area $q$. We therefore plotted against each other the logit ODR of laminar patterns for each pair of injected areas to their common targets, and fit a line of unit slope to the data in order to obtain the intercept (Fig. 3d). The absolute value of the y-intercept in each graph in Fig. 3d specifies a measure for hierarchical distance between the corresponding pair of injected areas (see 'Pairs plots' in Methods section)[18,19]. The y-intercepts indicate that V1 and POR occupy hierarchical levels that are most separated from all other areas (Fig. 3d, heat map). The majority of higher areas, however, show relatively low distance values in relation to each other, suggesting they occupy equivalent hierarchical levels. Moreover, in a strict hierarchy, the hierarchical distance between two target areas $p$ and $q$ should be independent of whether we

assess laminar patterns originating in area $i$ or those originating in another common source area $j$ and also terminating in $p$ and $q$[19]. Accordingly, we plotted against each other the logit ODR of pathways terminating in each pair of target areas and originating in all common source areas, and fit a line of unit slope to these plots (Supplementary Fig. 3c). These plots indicate that laminar patterns of projections terminating in V1 are the most dissimilar to patterns in all other areas, indicating V1 as being hierarchically distant from all other areas (Supplementary Fig. 3c heat map). LM and RL similarly appear to be relatively distant from most other areas when considering termination patterns of projections to these areas.

**Hierarchy of mouse visual areas**. We next used a beta regression model to estimate hierarchical levels for each area that best predicted the ODRs for each interareal pathway. Area V1 was set at level 0, and the model estimated the values for each area such that the difference between the hierarchical levels of any two areas best matched the logit ODR for the connection linking them. Figure 4a shows the estimated hierarchical levels for each area, and with the areas separated into dorsal and ventral stream branches[21]. The goodness of fit was evaluated by plotting the estimated hierarchical distances against the logit ODR values (Fig. 4b, $r = 0.85$, $t(78) = 14.30$, $p < 10^{-15}$; 95% conf. int.: (0.78, 0.90)). The plot demonstrates that hierarchical distances (i.e., the differences between hierarchical levels) are predictive of laminar projection patterns.

To identify the number of hierarchical levels that best describes the processing sequence, two or more areas were systematically constrained to be part of the same level by adding the columns from the incidence matrix that corresponded to these areas (described in Data analysis and statistics in Methods section), resulting in hierarchical models comprising fewer than 10 levels. A beta regression fit was performed for each such model, and the best model was assessed as the one with the lowest Akaike information criterion (AIC) value[8,27], which identifies the model with the best predictive power for the inclusion of new data. To generate the various models, a cumulative procedure was first employed in which the highest $n$ areas ($n$ going from 1 to 9), when arranged in the increasing order of their hierarchical levels from V1 to POR (Fig. 4a), were constrained to be part of the same level (Supplementary Fig. 4a). This was done by adding the next lowest column of the incidence matrix to the $n$ highest columns, starting with POR as the highest area, and performing the

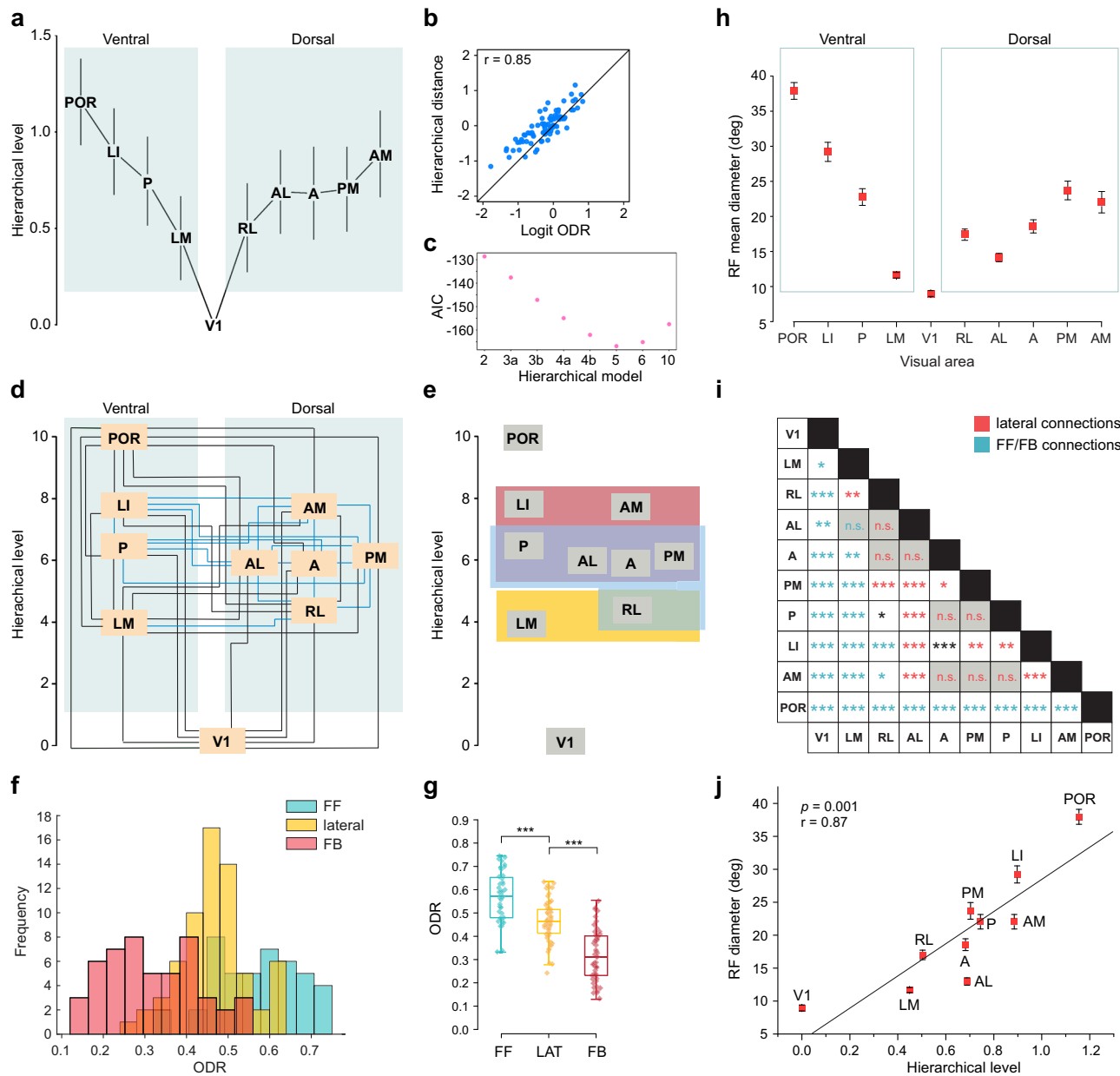

regression fit for each model. This generates a set of nested models. The model with the lowest AIC value among these nine cases was the one in which all areas were at different levels (Supplementary Fig. 4a, 'POR'). We next assessed another set of models, this time starting with LM and successively combining with it an increasing number of higher areas (starting with RL, and successively and cumulatively adding areas from A to LI) to generate eight different models (Supplementary Fig. 4b). Here, the hierarchy in which LM and RL were combined into one level, with all other areas at different levels, had the lowest AIC value, which was lower than each of the models in Supplementary Fig. 4a. Next, we looked at combinations of levels higher than LM/RL and lower than POR (Supplementary Fig. 4c). Finally, a model in which the hierarchy was composed of five levels (V1, LM/RL, A/AL/PM/P, AM/LI, and POR) had the lowest AIC value of all tested hierarchical models (Fig. 4c), making it the model that most effectively describes the network, balancing goodness of fit and model complexity.

Additionally, level values were estimated for each area relative to all other areas to evaluate whether the difference between the hierarchical levels reflected a significant separation. The organization of the areas based on such examinations is displayed in Fig. 4d in which black lines connect areas that are at significantly distant levels ($p < 0.05$, Wald test for multiple coefficients), and blue lines connect areas whose estimated level values lacked a significant difference; the latter were accordingly considered to indicate lateral connections because they interconnect areas that do not occupy different hierarchical levels. The results demonstrate that V1 and POR are significantly separated from all other visual areas in this network based on the beta regression model, whereas RL is the only area not hierarchically distant from LM (Fig. 4d, e). Furthermore, multiple higher-order areas could be grouped such that every areal pair within the group lacked a significant difference in their respective hierarchical positions (Fig. 4e). The hierarchy can therefore be thought of as comprising overlapping processing levels, each containing only lateral connections, i.e., pathways interconnecting areal pairs that are not hierarchically separated. In contrast to these nonhierarchical circuits, all connections emerging from or terminating in V1 and POR can be classified as being either FB or FF. Notably, despite

**Fig. 4 Construction of mouse visual cortical hierarchy. a** Estimated hierarchical levels obtained using a beta regression model such that the level value of V1 is set at 0, and differences between any two hierarchical level values best predict the ODR for pathways connecting the respective areas. Vertical lines demarcate 90% confidence intervals. The areas have been divided into previously described dorsal and ventral streams[21,25]. **b** Hierarchical distance values between all pairs of areas estimated by the beta regression model show a high goodness of fit with the logit of the measured ODRs ($r = 0.85$). **c** The Akaike information criterion (AIC) values for eight models in which different combinations of areas were constrained to be part of the same level, and the beta regression fit performed for each such model. The lowest AIC value occurs for the model with five levels (V1, LM/RL, A/AL/PM/P, LI/AM, and POR), indicating that this is the hierarchical model with the best predictive power. Hierarchical models: 2, 2 levels (all higher-order areas combined into one level, and V1 as a separate level); 3a, 3 levels (V1, LM, all higher areas merged into one level); 3b, 3 levels (V1, LM–LI, POR); 4a, 4 levels (V1, LM, RL–LI, POR); 4b, 4 levels (V1, LM/RL, A–LI, POR); 5, 5 levels (V1, LM/RL, A–P, LI/AM, POR); 6, 6 levels (V1, LM, RL, A–P, LI/AM, POR); and 10, all 10 areas considered as separate levels. **d** Hierarchical levels similar to Fig. 4a, but scaled to values between 1 and 10. Black lines interconnect pairs of areas that show a significant difference in their hierarchical levels ($p < 0.05$, two-sided Wald test for multiple coefficients). Blue lines interconnect areal pairs that lack a statistical significance in their hierarchical level. **e** Illustration of the overlapping hierarchy of the network. All pairs of areas within each colored box lack a statistically significant hierarchical separation, and pathways interconnecting these areas can therefore be considered to be lateral (i.e. neither FF nor FB). **f** Frequency distribution of ODRs for FF, lateral, and FB pathways. $p = 3 \times 10^{-24}$, one-way ANOVA; $n = 161$ laminar patterns from 20 injections. **g**. Box plots of ODR values for FF, lateral ('LAT'), and FB pathways. FF vs LAT, $p = 4 \times 10^{-7}$; LAT vs FB, $p < 2 \times 10^{-16}$; one-way ANOVA with post-hoc Tukey's range test; $n = 44$, 66, and 50 for FF, LAT, and FB pathways, respectively, from 20 injections. Box plots denote the median and are bound by the 25th and 75th percentile values, with whiskers denoting the 5th and 95th percentiles. **h** Receptive field diameters recorded in each area in anesthetized mice. Within each processing stream, RF diameters show an overall increase in areas at increasingly higher hierarchical levels for both dorsal and ventral streams ($p < 2 \times 10^{-16}$, one-way ANOVA; $n = 142$ and 164 neurons for the dorsal and ventral stream, respectively). This increase is more prominent in the ventral stream. Data are presented as mean values ± SEM. **i** Statistical significance of differences in RF diameters between all pairs of areas. *$p < 0.05$, **$p < 0.01$, ***$p < 0.001$, one-way ANOVA with post-hoc Tukey's range test. Gray blocks indicate no statistical significance (n.s.). Text (asterisks and n.s.) colors indicate whether the corresponding areas are connected by either lateral (red) or FF/FB (blue) pathways based on the anatomical hierarchy (Fig. 4d, e). Text in black indicates areal pairs that lack a connection in both directions. **j** Hierarchical level values are significantly correlated with RF diameters ($p = 0.001$, $r = 0.87$, Pearson's correlation; $n = 308$ neurons from 98 mice). Data are presented as mean values ± SEM.

the overlapping characteristic of the network, there exist multiple hierarchically organized processing routes, each comprising areas at significantly separated levels (Fig. 4d, e; for example, V1 → LM → LI → POR). This indicates the presence of multiple, hierarchical systems embedded within a network that exhibits nonhierarchical features.

We next examined the distribution of ODRs after classifying individual pathways into FF, lateral, and FB categories based on the hierarchical levels of the respective interconnected areal pair. FF, lateral, and FB connections were characterized by distinct distributions of ODR values (Fig. 4f, $p < 10^{-14}$, one-way ANOVA), with mean ODRs of $0.57 \pm 0.02$, $0.46 \pm 0.08$, and $0.32 \pm 0.11$, respectively (Fig. 4g, $p < 10^{-6}$ for FF vs. lateral, lateral vs. FB, and FF vs. FB, Tukey's range test). Despite the significant differences in the means of the ODRs, the overlap in ODR values between FF, lateral, and FB pathways indicate the presence of a hierarchy in which a subset of areas interact with each other in a nonhierarchical fashion.

**Receptive field sizes across the hierarchy.** Because areas at increasingly higher levels may be expected to integrate inputs from an increasingly larger number of lower areas with higher cortical magnification[28,29], an expected functional consequence of a hierarchy is that RF dimensions increase in higher-order areas[6]. To test if the hierarchy established on anatomical rules is consistent with the proposed summation of visual space, RFs were mapped for L2/3 neurons in each of the ten areas in response to drifting gratings in anesthetized mice ($n = 308$ neurons from 98 mice).

We separated the nine higher areas into the dorsal and ventral streams to examine if the RF diameters showed a successive increase along each stream. Mean RF diameters depended on both, hierarchical organization and processing stream (Fig. 4h). While both dorsal and ventral streams exhibited overall increases in RF diameters with increasing hierarchical positioning ($p < 0.001$ for dorsal stream, $p < 0.001$ for ventral stream, one-way ANOVA), the increase was more pronounced in the ventral stream (Fig. 4h), implying a stricter hierarchical organization in the ventral stream. Ventral stream areas LI and POR had the

largest RF diameters ($29.2 \pm 1.3°$, $n = 20$, and $38.0 \pm 1.1°$, $n = 20$, respectively) of all the areas, significantly larger ($p < 0.05$, Tukey's range test, Fig. 4h) than the RFs of each of the two highest dorsal stream areas AM ($22.0 \pm 1.1°$, $n = 18$) and PM ($23.7 \pm 1.3°$, $n = 24$; Fig. 4h, i). V1 ($8.9 \pm 0.5$, $n = 86$) and LM ($11.7 \pm 0.6$, $n = 72$) had the smallest mean RF diameters. We next examined if the structural hierarchy was consistent with differences in RF diameters. Whereas in the majority of cases, RF size differences were consistent with relative hierarchical positioning, with higher areas exhibiting larger RFs, we also found several instances of areas at the same hierarchical level exhibiting different RF diameters (Fig. 4i). A possible explanation for this observation is that while converging excitatory inputs from an increasingly greater number of areas contribute to larger RFs in higher areas, the hierarchical level of a particular area depends not only FF projections to that area but also on reciprocal FB projections and on FF projections to areas at even higher levels. Surround suppression and the distinct dendritic locations of different interareal inputs, both of which contribute to RF dimensions[29,30], were not accounted for in the hierarchical analysis. Nevertheless, RF diameters were significantly correlated with hierarchical levels ($p = 0.001$, $r = 0.87$, Pearson's correlation, Fig. 4j).

To further probe the robustness of the ODR-derived hierarchy, we examined its consistency with findings of previous anatomical and physiological studies based on much larger datasets. A recent study examined physiological properties, including the spike latencies and RF sizes, in each of six visual areas of head-fixed awake mice[6]. The 'hierarchy score' of these six areas, as determined by a clustering analysis of diverse anterogradely labeled laminar patterns[14], showed a strong correlation with the hierarchical level values of the present study (Supplementary Fig. 5a), indicating that the ODR is a reliable measure for hierarchical ordering. RF size (Supplementary Fig. 5b) and the time to first spike (Supplementary Fig. 5c) in L2/3 cells were significantly correlated with hierarchical level, with higher areas exhibiting larger RF sizes ($p = 0.005$, $r = 0.94$, Pearson's correlation) and longer latencies ($p = 0.02$, $r = 0.88$, Pearson's correlation) in their spike response in awake mice. We further performed pairwise comparisons of RF areas and spike latencies

to determine if areas at different hierarchical levels showed dissimilar physiological properties. Differences in RF sizes were consistent with hierarchical level differences in 11/15 cases (i.e., cortical areas at different hierarchical levels differed in their RF area, and those at the same hierarchical level showed no significant difference; Supplementary Fig. 5b, *t*-test with Holm-Bonferroni correction). For time to first spike, again 11/15 pairs showed a consistent relationship with hierarchical ordering (Supplementary Fig. 5c, *t*-test with Holm–Bonferroni correction).

**Hierarchy within each processing stream.** Dorsal and ventral streams are thought to underlie distinct functions[31–33]. We therefore asked whether analyzing the visual cortical system as two separate networks would result in a different hierarchical organization, and whether each stream exhibits a stronger hierarchical organization than the entire network of ten areas. For this purpose, we examined the hierarchical organization of the dorsal and ventral streams after the inclusion of only interareal connections within each stream. When connections between streams were eliminated, the negative correlation between the ODRs of reciprocal pathways was enhanced (Fig. 5a; slope = −0.53 when all pathways were included, slope = −0.90 when

only intra-stream pathways were included, $p = 0.009$ for comparison of slopes, Analysis of covariance), indicating a stricter adherence to FF/FB relationships within streams. The beta regression model was then used to construct the hierarchy of areas without cross-stream connections (Fig. 5b, c). The hierarchy showed modest changes compared with that in Fig. 4a in which all interareal connections were included. Notably, the level values of LI and P were further separated from LM and the network of dorsal stream areas, compared to the hierarchy diagram that included all interareal connections. Figure 5d shows the goodness of fit between hierarchical distance and logit ODRs after elimination of cross-stream pathways ($r = 0.90$, $t(49) = 14.52$, $p < 10^{-15}$; 95% conf. int.: (0.83, 0.94)). The best model was once again assessed using the AIC value. Here, a model comprising five levels – V1, RL/LM, AL/A/PM, AM/P, and LI/POR – had the lowest AIC value (Fig. 5e), indicating that such an organization provides for a model with the best predictive power when only intra-stream connections are considered.

We next asked whether the hierarchy would be maintained if we only examined regions that received the densest axonal projections, or whether the hierarchy is dependent on the use of specific axon density thresholds in L1 and L2-4. To do this, we repeated the analysis by including only pixels that had the highest

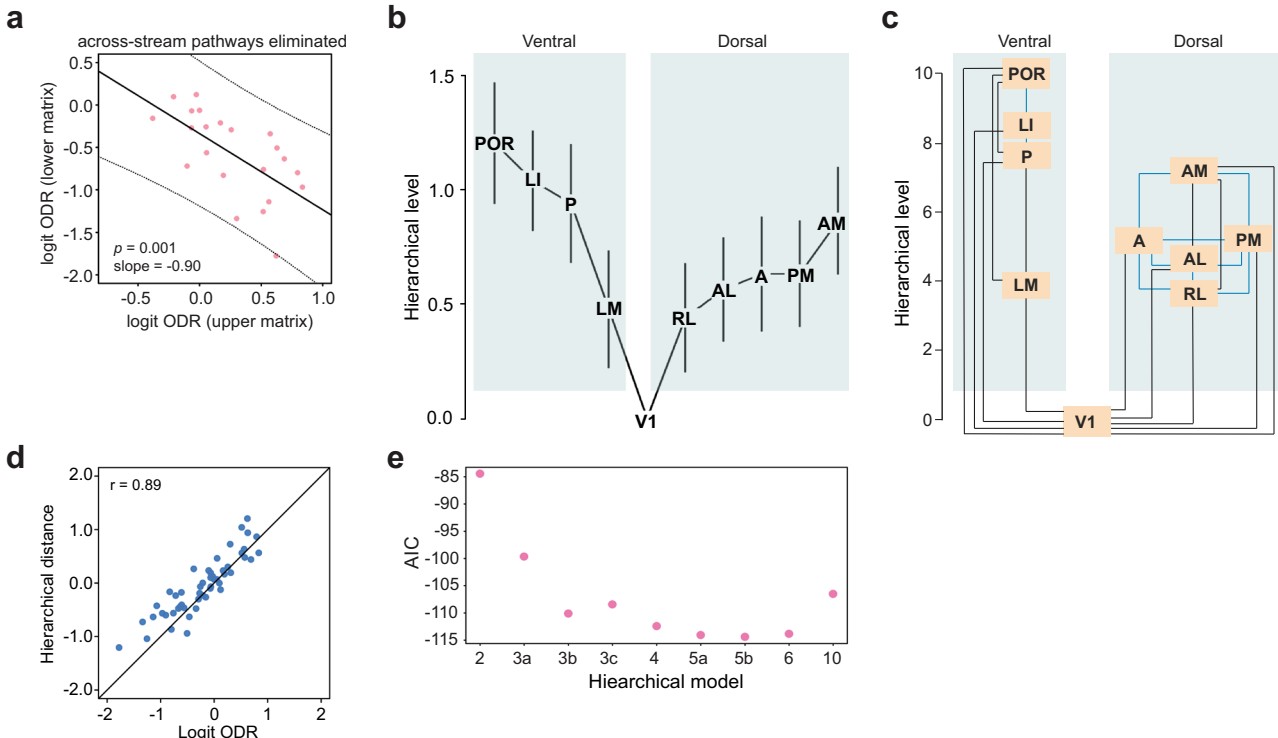

**Fig. 5 Construction of hierarchy after separation of dorsal and ventral streams. a** When pathways connecting areas belonging to the two different streams (dorsal and ventral) were eliminated from analysis, the plot of ODR values in one direction against that in the reciprocal direction shows a steeper slope compared to when all pathways are included (Fig. 3c; $p = 0.009$ for comparison between the two slopes, ANCOVA). **b** Estimated hierarchical levels obtained using a beta regression model after elimination of cross-stream pathways. Hierarchical level value of V1 was set at 0, and differences between any two hierarchical level values best predict the ODR for the pathways connecting the respective areas. **c** Hierarchical levels similar to Fig. 5b, but scaled to values between 1 and 10. Black lines interconnect pairs of areas that show a significant difference between their respective hierarchical levels ($p < 0.05$, two-sided Wald test for multiple coefficients). Blue lines interconnect areal pairs that lack a significant difference between their respective hierarchical levels. **d** Within-stream hierarchical distances estimated by the beta regression model plotted against the logit of the measured ODRs show a high goodness of fit ($r = 0.90$). **e** The AIC values for nine models in which different combinations of areas were constrained to be part of the same level, and the beta regression fit performed for each such model. The lowest AIC value occurs for the model in which the network is organized into five levels (V1, RL/LM, AL/A/PM, AM/P/LI, POR; model 5b), indicating this to be the model with the best predictive power. Hierarchical models: 2, two levels (V1, all higher-order areas merged into a second level); 3a, three levels (V1, RL/LM, all higher areas merged into a third level); 3b, three levels (V1, RL–PM, and AM–POR); 3c, three levels (V1, RL–LI, POR); 4, four levels (V1, RL–PM, AM/P/LI, POR); 5a, five levels (V1, RL/LM, AL/A/PM, AM/P/LI, POR); 6, six levels (V1, RL/LM, AL/A/PM, AM/P, LI, POR); and 10, all ten areas at different levels.

10% of pixel values and, alternatively, including only pixels within the highest 90% intensity values (representative examples in Supplementary Fig. 6a, b). The negative correlation between ODRs for reciprocally connected pathways was maintained in both cases (Supplementary Fig. 6c, d) indicating that this relationship is independent of axon density thresholds. The regression used for constructing the hierarchy resulted in a similar ordering of hierarchical levels for the 10 areas (Supplementary Fig. 6e–h) pointing to a robustness and invariance of the hierarchy across the percentile selection of projection densities within L1 and L2-4.

Additionally, to address the issue of whether the ODR at the center of each projection is significantly different from that at the edge of the projection, we repeated the analysis after elimination of pixels with the highest 10% values (Supplementary Fig. 6i, j). Inclusion of only the lowest 90% pixel values maintained the negative correlation of logit ODR values between reciprocal pathways (Supplementary Fig. 6i) and led to only modest changes to the hierarchy (Supplementary Fig. 6j).

## Discussion

Using anterograde pathway tracing, we have identified a graded parameter, the L2-4/L1-4 ODR, that is tightly correlated with a hierarchical distance measure, indicative of a hierarchical rule governing interareal connections in mouse visual cortex as has been reported in primate[17–19]. We show that, on average, the stronger the FF nature of a particular pathway, the stronger is the FB characteristic of the reciprocal connection. Unlike in the construction of earlier reported hierarchies[4,14,16], the approach used in the present study avoids a priori assignments of the FF or FB nature of individual projection patterns, except for projections emerging and terminating in V1. Our method made no attempt to minimize violations of the rule by which reciprocally connected pairs of areas interact through laminar patterns with opposite polarities[14]. Instead, we estimated level values for each area such that the difference between the levels of any two areas best predicted the laminar patterns of projections between them and to all their common targets. Thus, the ODR metric is neutral to violations of FF/FB relationships between areas. This approach allowed us to differentiate hierarchical from nonhierarchical relationships and to demonstrate that the network between higher visual areas contains hierarchical and nonhierarchical channels of communication. For example, in the hierarchical chain V1-RL-AM-POR, RL interacts nonhierarchically with AL, which is part of a parallel hierarchical chain V1-AL-POR. This design implies that the within-area local microcircuits of AL and RL may participate in lamina-specific FF/FB computations and, depending on demand, may be engaged by lateral inputs to most layers to dynamically influence the output of AL or RL.

The areal hierarchy described here exhibits differences with the recently published report by Harris et al.[14]. While both studies showed V1 at the bottom, and LM and RL at low levels of the hierarchy, we found that other areas, most notably LI, ranked near the top. Such disparities are attributable to the inclusion of projections to areas outside of the ten visual areas examined in this study, differences in the methodology for constructing the hierarchy, and the use of anatomical landmarks to identify areas in the present study as opposed to template matching to a common reference frame[14,34]. Notably, both studies point to a shallow visual hierarchy in the mouse, with fewer levels compared to the primate visual cortex[18]. The shallowness may reflect the larger RFs and the narrower range of RF sizes in mouse V1 than in primates. For example, it takes only four instead of eight steps up the hierarchy for pooling visual space into a 40° RF (Fig. 4h), as seen in the medial superior temporal area near the top of the

macaque cortical hierarchy[18,35]. The RF diameters of the L2/3 cells recorded in the present study were smaller than those of cells recorded in visual areas from awake head-fixed mice[6,36], although our results are consistent with the finding that RF sizes increase with increasing hierarchical levels. The RF dimensions of V1 neurons described here are comparable to those reported in previous studies in both anesthetized and awake mice[30,37].

The hierarchical ordering of RF sizes in the present study also differs from that in our previous report[23]. In the 2007 study, RF mapping was performed via a qualitative assessment of spike rate by listening to an audiomonitor, and plotting size and position manually on a spherical dome. Furthermore, high contrast bars and edges were used as stimuli to drive cells, which differ from the drifting gratings used in the present study. Thus, discrepancies between the findings of the two studies may be attributed to methodological differences.

The concept of a 'noncanonical' network has been previously used to describe the densely interconnected interareal circuit organization of primate association cortex as being distinct, structurally and functionally, from a strict hierarchy in which sensory signals are transformed to motor actions[15]. Noncanonical circuits, characterized by a lack of consistent hierarchical rules, were proposed to have expanded disproportionately during hominid evolution, and implicated in higher level cognitive capabilities such as planning and internal mentation. Such nonhierarchical networks may include numerous reciprocally connected areal pairs with pathways that resemble FF connections in both directions[19]. Our findings suggest that nonhierarchical circuits may not be a recent invention but already used at the front end for processing visual information. Notably, mouse visual cortex shares at least two fundamental features with primate association cortex. First, both networks comprise multiple, widely separated areas that receive common input from the pulvinar[32,38,39]. Such an organization may aid coordinating cortical areas to achieve specific visuomotor functions through motifs that are distinct from cortico-cortical processing schemes[38,40,41]. Second, pairs of association areas in primate cortex were shown to provide input to a common target through axonal projections that terminated in complementary laminar or columnar patterns[42]. Likewise, in addition to the diverse laminar patterns of projections to common target areas described here, projections from mouse visual areas LM, AL, and PM were recently shown to terminate in a modular interdigitating pattern in L1 of V1[43,44]. With L1 of areas LM, AL, LI, POR, and PORa exhibiting modular M2 muscarinic acetylcholine receptor expression patterns similar to that observed in V1[24,43], it is likely that these higher visual areas also receive complementary patterns of converging input from other pairs of areas within the network. Furthermore, higher visual areas have been implicated in the integration of visual signals with other modalities including vestibular, auditory, and somatosensory systems[45–48], and in the top-down control of context-dependent task performances[49,50]. Multisensory integration and the activity of higher-order cortical areas are thought to play key roles in decision-making and motor planning[49,51]. Taken together, these findings indicate that a nonhierarchical organization may be an evolutionarily ancient motif in the mammalian brain, operating in parallel with canonical hierarchies, and which become more emphasized in higher-order functions that are not directly involved in visual perception and oculomotor actions.

The ODR can be considered analogous to the percentage of supragranular labeled neurons (SLN), a distance measure that has been used to construct the primate cortical hierarchy[17–19]. Unlike the SLN, which is a measure of the proportion of retrogradely labeled neurons in L2/3 that project to a target area, a notable feature of the ODR is the inclusion of L1 connectivity in its

calculation. We find that L1, long believed to be the primary target of FB inputs providing modulatory and contextual signals[5,52], is a ubiquitous target of pathways interconnecting higher visual areas in the mouse, including those considered to be FF, FB, or lateral based on the hierarchical positioning of the corresponding areas, even though the relative axonal densities in L1 and L2-4 differed across these pathways. This indicates a critical role of L1, and therefore of inputs to the dendrites of the underlying neurons, in computations that do not exclusively involve "top-down" control of sensory processing. The prevalence of strong L1 inputs may contribute to the shallowness of the mouse visual hierarchy due to the paucity of prototypical FB pathways as described in the primate brain[4]. Mouse visual cortex also differs from monkey visual cortex with regard to FF projections, which are densest most commonly in L2/3 of the target area in the mouse but preferentially target L4 of that in the monkey[4]. Interareal axonal terminations in the mouse also show a prevalence of multilaminar patterns compared to monkey cortex, which more commonly exhibits unilaminar or bilaminar patterns[4].

The two visual processing streams are thought to underlie distinct functions[31,33] and to provide different temporal constraints to visual computations[6,53,54]. Here we show that the mouse ventral stream area LM and dorsal stream area RL reside at low levels in the hierarchy, with POR and AM at the highest levels of each stream. While dorsal stream areas AL, RL, and PM are all involved in processing optic flow information[55–57], they appear to underlie distinct functions. RL has been shown to be specialized for optic flow signals inherited from the retina[57,58] and for the encoding of proximal visual stimuli in the binocular field[59], whereas PM is not part of the specialized circuit that relies on direction selectivity produced in the retina[57,58]. RL and AL also form overall stronger connections than PM and AM with somatosensory areas, with PM and AM more strongly linked with cingulate and retrosplenial areas[8,21,48]. These findings suggest the existence of hierarchical substreams within the dorsal stream, analogous to that proposed in primate cortex:[60] we speculate that RL and AL are part of a posterior parietal substream that guides heading behavior and locomotion, and which relies on information acquired from direction selective retinal cells, and that PM and AM are linked more strongly with a parieto-premotor/prefrontal pathway that underlies the parsing of externally generated object movement relative to optic flow during self-motion.

Eliminating connections between the two streams enhanced the negative association between the ODRs of FF and FB pathways (Fig. 5a), consistent with the observation that cross-stream connections are generally 'lateral' with relatively few FF or FB pathways linking the two streams. Dense connectivity between the dorsal and ventral streams have been shown to exist in the primate[10], and are thought to underlie skilled visuomotor functions and color perception in humans[33,61]. These observations support the notion that nonhierarchical circuits expand the role of the visual cortex in mediating higher-order functions[10,15,62].

While our focus here was the intracortical visual network, the essential role of the thalamus in regulating visual signals must be considered in the description of cortical function. In addition to being involved in controlling interareal communication[40,41], the mouse pulvinar is also likely a critical component of signal processing in POR. POR was recently shown to be primarily activated not through the visual cortical hierarchy, but rather almost exclusively from signals transmitted through the collicular-pulvino-cortical pathway[63], further indicating that V1 is not the exclusive gateway into the cortical hierarchy, but that retinal signals can enter the cortical network at higher processing stages via difference sources including the pulvinar. Furthermore, because hierarchical relationships are often correlated with other

functional motifs, including response properties[6,29] and the balance between excitatory and inhibitory input[22,64], deducing the algorithms underlying hierarchical processing requires a detailed examination of thalamocortical and cortico-cortical connectivity at a cellular and dendritic resolution, as well as of the most efficient direct and indirect paths between areas for signal transmission across the network[10,21,65].

## Methods

All experimental procedures were approved by the Institutional Animal Care and Use Committee at Washington University in St. Louis (protocol no. 20190094).

**Animals**. In all, 6–16-weeks-old C57BL/6J male and female mice were used for the analyses of interareal axonal projection patterns. The in vivo single-unit recordings were performed in 5–8-weeks-old C57BL/6J male and female mice.

**Tracing axonal projections**. A single injection of anterograde tracer was performed in each animal. Each mouse was anesthetized using a mixture of ketamine and xylazine (86 mg kg$^{-1}$/13 mg kg$^{-1}$, IP), and thereafter secured in a head holder. To retrogradely label callosally projecting neurons for areal identification in the left hemisphere, 30–40 pressure injections (Picospritzer, Parker-Hannafin) of bisbenzimide (5% in H$_2$O, 20 nl each; Sigma) were made into the right hemisphere (occipital, temporal, and parietal cortices) using glass pipettes (20–25 μm tip diameter). Interareal connections within the left hemisphere were anterogradely labeled by inserting a glass pipette (15–20 μm tip diameter) into one of ten cortical areas and performing iontophoretic injections (3 μA, 7 s on/off cycle for 7 min; Midgard current source; Stoelting) of biotinylated dextran amine (BDA; 10,000 molecular weight, 5% in H$_2$O, Invitrogen). Injections of BDA were performed stereotaxically and at two depths (0.3 mm and 0.5 mm) below the pial surface. The stereotaxic coordinate system had its origin at the intersection of the midline and a perpendicular line drawn from the anterior edge of the transverse sinus at the posterior pole of the occipital cortex. The coordinates of the injected areas were (anterior/lateral in mm): V1, 1.1/2.8; LM, 1.4/4.0; AL, 2.4/3.7; RL, 2.8/3.3; PM, 1.9/1.6; P, 1.0/4.2; A, 3.4/2.4; LI, 1.45/4.2; AM, 3.0/1.7; POR, 1.15/4.3. Post-surgery analgesia was provided by subcutaneous injections of buprenorphine (0.05 mg kg$^{-1}$).

**Histology**. At least 3 days after BDA injections, the mice were overdosed with ketamine/xylazine (500 mg kg$^{-1}$/50 mg kg$^{-1}$, IP) and perfused through the heart with heparinized (0.01%) phosphate buffer (PB, 0.1 M, pH 7.4), followed by perfusion of 4% paraformaldehyde (PFA, in 0.1 M PB). Brains were removed, post-fixed (24 h, 4 °C), and equilibrated in sucrose (30% in 0.1 M, PB). To confirm the location of the injection site and to enable the identification of areal targets of axonal projections, bisbenzimide labeled neuronal landmarks in the left hemisphere were imaged in situ under a stereomicroscope (Leica MZ16F) equipped for UV fluorescence (excitation/barrier 360 nm/420 nm). The left hemisphere was then cut in the coronal plane on a freezing microtome at 40 μm. Sections were recorded as a complete series across the caudo-rostral extent of the hemisphere, and each wet-mounted section (in 0.1 M PB) was imaged under UV illumination using a fluorescence microscope equipped with a CCD camera (Photometrics CoolSNAP EZ, Sony) and MetaMorph software (Molecular Devices). BDA labeled fibers were visualized by incubating the sections in avidin and biotinylated HRP (Vectastain ABC Elite), and enzymatically reacted in the presence of diaminobenzidine and H$_2$O$_2$. Sections were mounted onto glass slides and cleared with xylene. The reaction product was intensified with AgNO$_3$ and HAuCl$_4$[21], coverslipped in DPX, and imaged under dark-field optics. Layers were assigned by the size and density of pale silhouettes of cell bodies whose distribution resembled the patterns seen in contrast-inverted Nissl stained sections.

**In vivo electrophysiology**. For single-unit recordings of receptive fields (RFs), mice were first anesthetized with urethane (20%, 0.2 ml/20 g body weight, i.p.). Tungsten microelectrodes dipped in DiI (5% in absolute ethanol) were inserted into each of the 10 visual areas in the left hemisphere, guided by stereotaxic coordinates. Recording depth was measured from the pial surface, and electrode insertion was controlled and monitored using a micromanipulator (Sutter Instruments). Recordings of spiking activity were acquired from L2/3 neurons using the TEMPO software (Tempo, Reflective Computing), and single units were isolated through the use of a digital spike discriminator (FHC Inc.). Recorded signals were amplified and bandpass filtered at 300–5000 Hz.

RF location was first determined by moving a light bar over a dark background of the monitor screen, and listening to the audiomonitor response to spike discharges. To measure RF sizes, a circular patch (5° diameter) of a drifting sinusoidal grating (0.03 c/deg) was presented at multiple locations on the display screen. Spatial response plots were generated from contour lines connecting points in visual space with similar mean response strengths to visual stimuli. The response strength for a neuron was measured as the mean firing rate during the 2-s stimulus. For each recorded neuron, the response field was fit with a Gaussian, and the RF diameter was computed from the contour corresponding to two standard

deviations (SDs) of the fitted Gaussian after transforming the elliptical field into a circle. Average RF diameters are presented as means ± standard error of the mean.

*Confirmation of recording site.* After each recording, mice were perfused through the heart with heparinized PB followed by 1% PFA (see Histology for details). The left cortical hemisphere was separated from the rest of the brain, flat-mounted, postfixed in 4% PFA, and cryoprotected in 30% sucrose[43]. The flattened cortices were sectioned at 40 μm in the tangential plane on a freezing microtome. The sections were washed with 0.1 M PB, treated with 0.1% Triton X-100 and normal goat serum (10% NGS in PBS), and incubated with an antibody against the M2 muscarinic acetylcholine receptor (1:500 in PBS, MAB367, Millipore) for at least 24 h. Next, sections were washed with 0.1 M PB, and treated with a secondary antibody labeled with Alexa Fluor 647 (1:500 in PBS, Invitrogen). Sections mounted on glass slides were imaged with a CCD camera (Photometrics Cool-SNAP EZ), and the location of the recording site marked with DiI was compared with the M2 staining pattern. The areal location of the recording site was assessed using published maps based on M2 expression[8,43].

**Data analysis, statistics, and reproducibility**. Two animals were used for the injection of each area; thus a total of 20 injections (one injection per mouse) were performed. For each injected animal, 3–5 adjacent coronal sections containing anterogradely BDA labeled projections in each target area were used for quantification of termination patterns. Occasionally, the terminal projections contained retrogradely labeled cells. Such projections were excluded from analysis. Projections were assigned to areas by their location relative to bisbenzimide-labeled callosal landmarks[22,23], and based on their locations relative to other projection sites. BDA labeled projection fields were then superimposed over the pattern of bisbenzimide labeled callosal patterns in the same section, and axonal terminations were assigned to specific areas according to previously published maps[8,23].

*Optical density ratio.* For analyses of the termination pattern in each area, each coronal section was imaged under dark-field illumination at ×10 magnification, after which grayscale images were recorded. To quantify the laminar termination patterns formed by each interareal pathway, pixel intensity values from layer 1 (L1) and from layers 2 to 4 (L2-4) were averaged to obtain the ratio of the optical density of labeled axons in L2-4 to that of the total axonal projections in L1 + L2-4. This ratio, referred to as the ODR, was measured in each coronal section, and averaged for each pathway. Pixel intensity values were obtained from the grayscale images using a custom-written MATLAB script. Regions in L1 and L2-4 that lacked labeled axons were used to calculate background intensities in each section. These background intensity values were respectively subtracted from all pixel values in L1 and L2-4. To avoid oversampling of background regions, we only selected pixels within the top 70% of the highest intensity value in each section in Figs. 2–5, and pixels either within the highest 10% or the highest 90% in Supplementary Fig. 6. A 1-μm radius smoothing function was performed on each grayscale image before averaging the respective pixel values in L1 and L2-4 to avoid the overrepresentation of outlier and saturated pixels. Average ODRs are presented as means ± standard error of the mean. Box plots of ODRs denote the median and are delineated by the 25th and 75th percentile values, with whiskers denoting the 5th and 95th percentiles.

*Pairs plots.* We evaluated the consistency of hierarchical relationships across the network of visual areas by identifying a hierarchical distance measure that would be independent of the injection site and target areas (Fig. 3d and Supplementary Fig. 3c). Formally, we are looking for a hierarchical distance value $\delta_{ij}$ between any two areas i and j, which we define as the difference between the respective hierarchical level values ($h_i$ and $h_j$) of the two areas. Thus,

$$\delta_{ij} = h_i - h_j$$

For two cortical areas p and q that are each injected independently, we can estimate the hierarchical distance measure from each of these areas to a common target area i:

$$\delta_{ip} = h_i - h_p$$

$$\delta_{iq} = h_i - h_q$$

Therefore,

$$\delta_{ip} - \delta_{iq} = (h_i - h_p) - (h_i - h_q) = h_q - h_p = \delta_{qp}$$

$$\delta_{ip} = \delta_{iq} + \delta_{qp}$$

where $\delta_{qp}$ is the estimated hierarchical distance between areas q and p. This relation is in the form of a linear function for a line of unit slope in which the intercept gives the hierarchical distance between the two injected areas. Thus, for any target area i, a hierarchical distance measure between two injected areas can be estimated by plotting against each other the anatomical indices (such as the logit transformation of the ODR) for the respective hierarchical distances between the two injected areas and the shared target area i. Accordingly, the y-intercept in each

panel of Fig. 3d provides a measure of hierarchical distance for the corresponding pair of injected areas.

*Estimation of hierarchical levels.* Statistical tests for generating and analyzing the cortical hierarchy were performed in the Open Source software R[66,67]. The regression analysis used here is an adaptation of the previously reported model that used discrete counts of retrogradely labeled cells instead of the continuous measure of optical densities of anterogradely labeled fibers used here for quantifying hierarchical relationships[18,19].

We used beta regression to estimate hierarchical distance values between a pair of areas that would best predict the ODR for pathways connecting the two areas. The beta distribution is a continuous probability distribution defined on the interval (0, 1) typically parameterized by two shape parameters, α and β, with a probability density function

$$f(x;\alpha,\beta) = \frac{x^{\alpha-1}(1-x)^{\beta-1}}{B(\alpha,\beta)},$$

where $B(\alpha,\beta) = \frac{\Gamma(\alpha)\Gamma(\beta)}{\Gamma(\alpha+\beta)}$, where Γ is the Gamma function. In the betareg package[66], the distribution is reparameterized in terms of the mean, $\mu = \frac{\alpha}{\alpha+\beta}$, and a precision parameter, $\phi = \alpha + \beta$, with probability density

$$f(x;\mu,\phi) = \frac{\Gamma(\phi)}{\Gamma(\mu\phi)\Gamma((1-\mu)\phi)} x^{\mu\phi-1}(1-x)^{(1-\mu)\phi-1},\ 0 < x < 1,$$

where $0 < \mu < 1$ and $\phi > 0$.

To fit a hierarchical model, we estimate the expected value of the ODR and set the logit transformation of the ODR equal to the hierarchical distance between two areas. This defines a relation between a linear predictor and the ODR

$$\text{logit}(\text{ODR}) = X\delta,$$

where $X$ is the incidence matrix for the graph of areal connections, i.e., each column corresponds to one of the 10 areas and each row corresponds to a link between a pair of areas. Each row is composed of 0's except for the columns corresponding to the connection between the two areas, taking the values −1 and 1, depending on whether they are the source or target. The vector δ contains the estimated hierarchical levels assigned to each area (column). All of the rows sum to zero. Therefore, in order to yield an identifiable solution, one column is dropped – here the column corresponding to area V1 – and its hierarchical level is, thereby, fixed at 0. Provided the incidence matrix and the vector of ODR values, the betareg function returns maximum likelihood estimates of the vector δ and the precision parameter, φ and their standard errors estimated from the variance-covariance matrix. The solution is unique only up to addition of a constant and/or multiplication by a factor. The beta regression was calculated using the *betareg* function in the betareg package (ver. 3.1-4)[66].

The Akaike information criterion (AIC)[27] was used to assess the number of levels that best described the hierarchy. The AIC is defined as minus twice the log likelihood plus twice the number of parameters estimated. In the large sample limit, it approximates the same result as the leave-out-one cross-validation score, thus giving a measure related to prediction error. The model with the lowest AIC will yield the best balance of goodness of fit and model complexity and among a set of models is expected to best predict new data sets. The AIC values for the different models (see Results section) were obtained using standard methods in R.

**Reporting summary**. Further information on research design is available in the Nature Research Reporting Summary linked to this article.

## Data availability
Logit ODR data for all pathways are available at github.com/rdsouza2882/ NatComm2021. Additional data that support the findings of this study are available from the corresponding author (A.B.) upon reasonable request. Source data are provided with this paper.

## Code availability
Custom-written code used to analyze data is available at github.com/rdsouza2882/ NatComm2021.

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

## Acknowledgements

This work was supported by the National Eye Institute (R01 EY016184, R01 EY020523, R01 EY022090, and R01 EY027383 to A.B.), and LABEX CORTEX (ANR-11-LABX-0042 to H.K.) of Université de Lyon (ANR-11-IDEX-0007) operated by the French National Research Agency (ANR), (ANR-15-CE32-0016 CORNET to H.K.), (ANR-17- NEUC-0004 to H.K.), (A2P2MC to H.K.), (ANR-17-HBPR- 0003, CORTICITY to H.K.), and (ANR-19-CE37-0000, DUAL_STREAMS to K.K.).

## Author contributions

Q.W. and A.B. designed experiments; Q.W., R.D.D., W.J., and A.B. performed experiments; R.D.D., K.K., A.M., and W.J. analyzed data; R.D.D., A.B., K.K., and H.K. wrote the manuscript

## Competing interests

The authors declare no competing interests.
