## [Peer Review File · Nature Communications]

Hierarchical and nonhierarchical features of the mouse visual cortical networkREVIEWER COMMENTS

Reviewer #1 (Remarks to the Author):

This wonderful and important study shows that there is a clear logic to the connections between visual areas in the mouse cortex. The background of the study is that the laminar pattern of axons sent from one area to another displays typical features if it is a feedforward connection (axons avoid layer 1) or a feedback connection (axons favor layer 1). Based on this fact, the study reveals a remarkable finding: that the more "feedforward-looking" the connection in one direction, the more "feedback-looking" the connection in the other direction. Moreover, this internal logic to the connections allows the authors to build a hierarchy of visual cortical areas in the mouse that is in many ways as compelling as similar hierarchies that have been drawn for primates. The paper also has weaknesses, which must be addressed but are all easy to address with simple revisions. The rest of this review will focus on these weaknesses, but to be clear: the overall impression of this study, as explained in the paragraph above, is extremely favorable. This is an important study.

MAJOR

The index used to measure optical density ratio (ODR) is inadequate, and indeed the paper spends considerable and poorly-spent effort tweaking it. The index is first defined as $L1/L24$, where $L1$ is density in layer 1 and $L24$ is density in layers 2-4. This is clearly a bad choice for two reasons: (1) the index will go from zero (when numerator is zero) to infinity (when denominator is zero); (2) we have no idea whether the variation between the two measures is substantial or small relative to the overall signal. Indeed, the first thing the paper does is to take a logarithm of it. This is the obvious choice for a ratio, where the value of 2 is as distant from the value of 1 as the value of $1/2$. But this takes care only of problem 1: we still have no idea whether the variation between the two measures is substantial or small relative to the overall signal. By the time we are at Figure 3 the paper has realized this, and we are now given the more logical measure " $ODR/(1+ODR)$ ", which is simply $L1/(L1+L24)$. Now things have improved. This index goes from 0 to 1 so we no longer have the problem of numbers going to infinity (and indeed the distribution looks bell-shaped, without recourse to logarithms). Solution to this problem: use a more logical index than ODR, from the very beginning of the paper. One possibility is to use the index that was used when making Figure 3, i.e. $L1/(L1+L24)$. This index could be simply described in plain English words: "relative density in layer 1". Another possibility is to use the method that is commonly used when measuring all sorts of indices: the differences divided by the sums, i.e. $(L1-L24)/(L1+L24)$. This index could be called "projection density index" and would also do a good job.

The physiology data are not nearly of the same quality as the anatomical data, because they were acquired under anesthesia. This is no longer state of the art in the field, because under anesthesia most of the circuitry is shut down and inhibition behaves differently. Even in wakefulness there are differences due to arousal, but that's a detail compared to how massively different (and delayed) the responses are during anesthesia in mouse visual cortex. At this point, in 2020, measurements obtained under anesthesia in mouse visual cortex do not belong in a top journal. This is a bit of a pity because if the measurements were valid (done in wakefulness), one could use them for interesting measurements such as measuring contrast responses, which in primates show a clear pattern as one progresses along the visual pathway (Sclar, Maunsell, and Lennie, 1990, and related work in humans), and to test for the a progressive increase in response delay (work by the Bullier laboratory). These analyses are not possible with data obtained under anesthesia in mouse. The suggestion therefore is to minimize the role of the physiological data in this paper. Drop the examples in Figure 4a, which are not nearly of the same quality as other data that has appeared in the literature, and consider putting Figures 4b and c in the supplementary materials. This will not diminish the quality of the paper, which is largely an anatomical study (and a wonderful one).

Figure 4. The analysis here assumes that the visual areas divide into two streams (ventral and dorsal). But this is based on previous studies. Please explain whether the present data and the present analysis support this division in streams? This is an important point and is presently completely unclear in the paper.

SPECIFICS

The emphasis on “canonical” and “noncanonical” seems a bit odd. These words are used in some specialized papers, and most readers will have no idea what they mean. Why not use simpler concepts that actually carry meaning, such as “hierarchical” vs. “parallel”.

Lines 43-44. When describing the processes supported by feedforward circuits, consider citing the work of the DiCarlo laboratory, which reveals how object recognition in primates happens in a feedforward fashion in the first 100 ms of vision.

Line 46. When describing the processes supported by feedback circuits, there are much stronger examples than the speculative and weak papers arguing for “predictive coding”. Processes that are solid and established include attention, figure-ground segregation (e.g. Roelfsema et al. Neuron 2007), and more.

Lines 207-213. It's fine to fit a gamma distribution to the data, but (1) we don't need the equation for a gamma distribution, it's in the textbooks already; (2) if we are going to use the values of the parameters (are we? It's not clear that the paper actually uses them) then we need to see fits of the function, to judge if they are good or not. In fact, it is not clear that we need the gamma distribution at all in this paper.

Line 230. This is the apex of oddity with the current definition of the ODR. We are going to take the exponential of the logarithm of it? To end up with a ratio of ODRs? Please define a reasonable index as explained above, and drop all this business of exponentials of logarithms because it is at the same time trivial and odd.

Lines 243-271. This first paragraph of Results directs us to look at Figure 1a, Figure 1b, and Figure 1c, meaning we are supposed to look at 27 panels, each with 6 layers. At the same time, we are meant to look at a supplementary figure and we are pointed to 8 more supplementary figures. We are barely told what to look for in these figures. The result is completely overwhelming. Please limit the calls to figures to the minimum necessary in this paragraph, and let the next paragraph do its job (which it does well), namely to guide the reader in the comprehension of feedforward circuits (figure 1a) and thence to the rest.

Figure 1. This will sound like a drastic suggestion but hear it out: consider inverting the color of the anatomical images. This reviewer tried this inversion to check that it looks good, and it does look very good. The main advantage is that one does not have large black squares, which have high contrast relative to the white page but carry no information.

Figure 1. Next to each anatomical image, please provide the index computed for that specific image (which has to be something better than ODR, as explained above).

Line 277. “encompassed by L2-4”. In those images one sees activity throughout cortex with only exceptions being L1 and perhaps a hole in L4 (is it significant? It seems to be but this is not explained in the paper). So, it is not clear why this sentence focuses on L2-4 (as a reader, one is confused). Better to first point out the wide expression before L1 and then go in the details.

Figure 2a. It's clear why there are no values in the diagonal but: could there be values perhaps? It would be interesting to know what the values of the index are for intra-areal connections.

Lines 319-320 this expectation (“if a pathway is feedforward in one direction, the reciprocal...”) is actually confirmed by the data, and this is one of the strongest points of this paper. Consider emphasizing this in abstract, introduction, etcetera.

Figure 2. The scatter plots in Fig 2c are too small and numerous to follow. Please combined them into a scatter plot with all the data, combined, showing one dot per connection. Perhaps color the plots according to the areas of origin and of destination (for instance, the fill color could indicate the area of origin, and the outline color could indicate the area of destination). If this ends up being confusing, break it up into a few scatter plots. For instance, one scatter plot for the first

column of 2c, one for the next column, and so on (that's 9 scatter plots instead of 45 scatter plots).

Figure 2c, rescale the axes to have equal ranges, so that the identity line is at 45 degrees. Lines 343-344. These two equations are unnecessary (by adding equations for things that are trivial, one risks losing readers).

Line 366. The "inverse logistic transformation of the natural log of the ODR" is a sentence that perfectly illustrates the strangeness of the approach, and so is the equation below it. Please just choose a reasonable index to begin with, as described above, and stick to it throughout the paper. It is also not clear at all that we need to hear about beta distributions. Where are the fits of the distribution to the data? What are we doing with the parameters of the fit? If they are not needed, please drop the whole concept. Just use a reasonable index and show that its distribution is bell-shaped.

Figure 3. By the time we have reached this figure we have seen ODR, $\log(\text{ODR})$, $\text{ODR}/(1+\text{ODR})$, and its (more understandable) synonym $L2-4/(L1-L2-4)$. The latter two appear in the same figure. This is not a good idea. Please choose a reasonable index from the outset, and stick to it.

The Discussion (and to some extent also the Introduction) is verbose. Consider trimming it.

The supplementary figures need headings and labels and could be streamlined and combined. For instance, one could put all the connections from one area to another in a column, and put multiple destination areas on the same page (e.g. 9 rows and 5 columns in one page, and 9 rows and the remaining 4 columns in the next page). This would not only reduce the number of supplementary figures but also make them easier to navigate.

Reviewer #2 (Remarks to the Author):

D'Souza, et al. investigated hierarchical organization of mouse visual cortex. The study performed tracer injections in all the visual areas to determine the distributions of the axon terminals across layers in the target areas. Based on the axonal projection patterns, the authors proposed a hierarchical relationship among the visual areas. Their electrophysiological recordings show that the areas at higher hierarchical levels show increasingly larger receptive fields.

This study tackles a long-standing question in cortical neuroscience, how distinct cortical areas are organized. The mouse visual cortex now provides one of the most popular models. For example, a recent paper from the Allen Institute addressed a similar question (Harris et al 2019, Nature) based on one thousand tracing experiments and clustering analyses. A manuscript at bioRxiv from the same institute shows functional correlates of the anatomical hierarchy based on open-source large datasets. The current study investigates this topic with smaller datasets using tracer injections and a small number of single unit recordings from anesthetized mice. Although the topic is important, I have the following major concerns. With its limited analyses and datasets, the manuscript is of interest to limited specialists, and not to the general readers of Nature Communications.

1. As one of the main conclusions of this study, the authors proposed an overlapping hierarchy levels (V1, LM, RL and the rest) with canonical and non-canonical network motifs. This proposal is of potential interest, but readers are left without any actual support. The study showed how the receptive field size changes in the hierarchical organization, and how it can be captured by the ventral and dorsal streams. This can be consistent with canonical motifs. However, the relation between this size change and the four hierarchy levels proposed in this study is not clear. Nor is clear its relation with the lateral connections. The proposal on the non-canonical network remains as a hypothesis that cannot be tested.

2. Their physiological measurements address an important question, how visual representation is transformed along the hierarchical ladder. However, there are three concerns. First, it is not clear why the findings here are not consistent with the report from the same lab (Wang and Burkhalter 2007, JCN). In their previous report, the order of the receptive field sizes were $V1 < LM < POR < AL < P < LI < PM < AM < RL < A$. How do the study reconcile these differences? Second,

there are many other receptive field parameters to compare, for instance, latency, linearity, etc. Those parameters should be reported in their relationship with the hierarchical organization. Third, the data sample number is too small ($n < 10$) in each area (except for V1 and LM). $N = 10$ cannot be sufficient, considering the possibility of layer and cell-type differences in a given area.

Therefore, although the physiological data are interesting preliminary findings, I would expect more complete datasets and analyses for publication in Nature Communications.

Other minor concerns are the following

1. Lateral connections.

Lateral connections are the key to this study. The connections, however, are basically an unclassified group, that cannot be identified as either feedforward or feedback. As mentioned in the discussion, the connections can include reciprocally connected pairs of areas, exhibiting feedforward and feedback termination patterns in both directions. The study should prepare a figure or supplemental figure that shows various types of termination patterns in both directions.

2. Difference between this study and Allen Institute's report in 2019

Allen Institute uses different categorization of feedforward and feedback projections for laminar distributions. In addition, Allen's report included thalamo-cortical, and cortico-thalamic projections to define their hierarchy. Please clarify the advantage and disadvantage of the current study over the one by Allen Institute.

3. Densest 70% of projections

The conclusions of the study is built on the measurement of average optical density of projections. The study uses the densest 70% of projections. It is important to show whether the conclusions remains the same with different criterion (e.g., 90%, 50%).

4. Please consider a schema of the injection site and target sites for figures.

It will be helpful for the general readers to prepare a schema for each anatomical experiment as supplementary figures.

Point-by-point responses to the reviewers' comments

Reviewer #1

- 1.1 The index used to measure optical density ratio (ODR) is inadequate, and indeed the paper spends considerable and poorly-spent effort tweaking it. The index is first defined as $L1/L24$, where $L1$ is density in layer 1 and $L24$ is density in layers 2-4. This is clearly a bad choice for two reasons: (1) the index will go from zero (when numerator is zero) to infinity (when denominator is zero); (2) we have no idea whether the variation between the two measures is substantial or small relative to the overall signal. Indeed, the first thing the paper does is to take a logarithm of it. This is the obvious choice for a ratio, where the value of 2 is as distant from the value of 1 as the value of $1/2$. But this takes care only of problem 1: we still have no idea whether the variation between the two measures is substantial or small relative to the overall signal. By the time we are at Figure 3 the paper has realized this, and we are now given the more logical measure “ $ODR/(1+ODR)$ ”, which is simply $L1/(L1+L24)$. Now things have improved. This index goes from 0 to 1 so we no longer have the problem of numbers going to infinity (and indeed the distribution looks bell-shaped, without recourse to logarithms). Solution to this problem: use a more logical index than ODR, from the very beginning of the paper. One possibility is to use the index that was used when making Figure 3, i.e. $L1/(L1+L24)$. This index could be simply described in plain English words: “relative density in layer 1”. Another possibility is to use the method that is commonly used when measuring all sorts of indices: the differences divided by the sums, i.e. $(L1-L24)/(L1+L24)$. This index could be called “projection density index” and would also do a good job.

We agree with the reviewer that our initial approach was unnecessarily complicated. We have redefined the ODR as $L2-4/(L1 + L2-4)$ and have revised all of the analysis of the anatomy data. Please also see our responses to this issue in the SPECIFICS portion of the reviewer's comments (sections 1.8, 1.18, and 1.19).

- 1.2 The physiology data are not nearly of the same quality as the anatomical data, because they were acquired under anesthesia. This is no longer state of the art in the field, because under anesthesia most of the circuitry is shut down and inhibition behaves differently. Even in wakefulness there are differences due to arousal, but that's a detail compared to how massively different (and delayed) the responses are during anesthesia in mouse visual cortex. At this point, in 2020, measurements obtained under anesthesia in mouse visual cortex do not belong in a top journal. This is a bit of a pity because if the measurements were valid (done in wakefulness), one could use them for interesting measurements such as measuring contrast responses, which in primates show a clear pattern as one progresses along the visual pathway (Sclar, Maunsell, and Lennie, 1990, and related work in humans), and to test for the a progressive increase in response delay (work by the Bullier laboratory).

These analyses are not possible with data obtained under anesthesia in mouse. The suggestion therefore is to minimize the role of the physiological data in this paper. Drop the examples in Figure 4a, which are not nearly of the same quality as other data that has appeared in the literature, and consider putting Figures 4b and c in the supplementary materials. This will not diminish the quality of the paper, which is largely an anatomical study (and a wonderful one).

The reviewer expresses concerns that measuring RF sizes in anesthetized mice misrepresents their true dimensions. We are aware that anesthesia reduces suppression from the RF surround and increases the RF center of V1 neurons (Vaiceliunaite et al., 2013; Self et al., 2014). However, whether RF size in higher visual areas is affected by anesthesia is unknown. It is important to note that the RF sizes we have found in L2/3 of V1 closely match those reported for “wide” and “narrow” spiking cells by Niell and Stryker (2008) in anesthetized mice. In contrast, Siegle et al. (2021) found in awake headfixed mice that the mean RF sizes across all layers of V1 are substantially larger. This is in sharp contrast to RFs in L2/3 of V1 of awake mice (Keller et al., 2021), which are a modest 3-4 degrees larger than the RFs we have recorded in L2/3 under urethane anesthesia. Thus, while anesthesia clearly affects RF size, the variability is most strongly driven by the 3-fold size difference between supragranular and infragranular layers (Niell and Stryker, 2008). We would also like to point out that Siegle et al. (2021) used a 20 deg. circular Gabor patch as their receptive field mapping stimulus, making it unlikely for them to be able to record RFs smaller than 20 deg., which possibly explains their relatively large RFs.

Although the absolute RF sizes in higher visual areas differs between our study and that of Siegle et al. (2021), it is important to note that in both studies RFs in PM and AM are significantly larger than in V1, LM, and RL, consistent with their rankings at different hierarchical levels. While the rankings by RF size and hierarchical distance are not perfectly matched (e.g., RL and PM differ by RF size but are structurally at the same level), 8 of 9 pairs of areas in which RFs are not significantly different in size are ranked at similar hierarchical levels (Fig. 3g). Thus pairwise comparisons of RF size and a structural metric based on ODRs of shared projection targets show a remarkably similar hierarchical ranking of visual areas.

We considered putting the physiological findings into a supplementary figure, but we saw an advantage in including it in Fig. 3 where comparisons between anatomical and physiological hierarchies, which show remarkable similarities, could be effectively made.

References:

Vaiceliunaite A et al. (2013) Spatial integration in mouse primary visual cortex. *J Neurophysiol* 110(4): 964–72.

Siegle JH et al. (2021) Survey of spiking in the mouse visual system reveals functional hierarchy. *Nature* doi: 10.1038/s41586-020-03171-x

Self MW et al. (2014) Orientation-tuned surround suppression in mouse visual cortex. *J Neurosci* 34(28):9290-304. doi: 10.1523/JNEUROSCI.5051-13.2014

Keller AJ et al. (2020) Feedback generates a second receptive field in neurons of the visual cortex. *Nature* 582(7813):545-549. doi: 10.1038/s41586-020-2319-4

Niell CM and Stryker MP (2008) Highly selective receptive fields in mouse visual cortex. *J Neurosci* 28: 7520–7536.

- 1.3 **Figure 4. The analysis here assumes that the visual areas divide into two streams (ventral and dorsal). But this is based on previous studies. Please explain whether the present data and the present analysis support this division in streams? This is an important point and is presently completely unclear in the paper.**

The reviewer has correctly noted that the division of streams was based on previous studies showing preferential connection weights within each stream. To address the issue of whether the hierarchical organization is compatible with the notion of streams, we repeated the analyses after including only connections between areas within the same stream. The results indicate that the majority of cross-stream connections are “lateral” in nature so that excluding them leads to a steeper hierarchy in the ventral stream, and a stronger adherence to the rule that the more feedforward (FF) a pathway is in one direction, the more feedback (FB) it is in the reciprocal direction. Accordingly, we have included a new figure, Fig. 4, which addresses how separating the two streams affects the hierarchy.

SPECIFICS

- 1.4 **The emphasis on “canonical” and “noncanonical” seems a bit odd. These words are used in some specialized papers, and most readers will have no idea what they mean. Why not use simpler concepts that actually carry meaning, such as “hierarchical” vs. “parallel”.**

We considered the reviewer’s suggestion in replacing “noncanonical” with “parallel” or “lateral”, but found that both terms were used in descriptions of interareal circuits and lacked a clear reference to a hierarchical organization, which is more clearly conveyed by the term “noncanonical”.

A central finding of our study is that the mouse visual cortical network comprises both hierarchical and non-hierarchical features. Networks that are distinct from sensory-motor hierarchies have been previously described as being “noncanonical”. Such noncanonical networks have been implicated in higher-order cognition in hominids and other primates, and we find them to be a notable feature of mouse cortex. We believe the presence of noncanonical networks, which appears to be an evolutionarily conserved motif, is an important finding, and have therefore elaborated in more detail the concepts of canonical

and noncanonical networks in the Introduction and the Discussion sections (lines 49–55, 358–367).

- 1.5 Lines 43-44. When describing the processes supported by feedforward circuits, consider citing the work of the DiCarlo laboratory, which reveals how object recognition in primates happens in a feedforward fashion in the first 100 ms of vision.

We have now cited a study by DiCarlo et al. (2012) that investigated the role of the FF pathway in object recognition in the Introduction (line 47).

- 1.6 When describing the processes supported by feedback circuits, there are much stronger examples than the speculative and weak papers arguing for “predictive coding”. Processes that are solid and established include attention, figure-ground segregation (e.g. Roelfsema et al. Neuron 2007), and more.

We agree with the reviewer that our focus on predictive coding in the original version of the manuscript was arguably unsuitable with regard to the aims of the study. Accordingly, we have revised the Introduction and have cited two studies by the Roelfsema group (Self et al., 2013; Vangeneugden et al., 2019) both in the Introduction (line 43) and the Discussion (lines 381, 392).

- 1.7 It’s fine to fit a gamma distribution to the data, but (1) we don’t need the equation for a gamma distribution, it’s in the textbooks already; (2) if we are going to use the values of the parameters (are we? It’s not clear that the paper actually uses them) then we need to see fits of the function, to judge if they are good or not. In fact, it is not clear that we need the gamma distribution at all in this paper.

We agree that defining the beta distribution in the main text is unnecessary, and have accordingly revised the Results section. The primary objective here was to use a regression method to fit data that resembles a proportion or probability (with values ranging between 0 and 1). We have clarified this point in the Methods (lines 570–574).

- 1.8 This is the apex of oddity with the current definition of the ODR. We are going to take the exponential of the logarithm of it? To end up with a ratio of ODRs? Please define a reasonable index as explained above, and drop all this business of exponentials of logarithms because it is at the same time trivial and odd.

We have reanalyzed all of the anatomy data by using a new definition of the ODR index: $L2-4/(L1 + L2-4)$, which necessarily only includes values between 0 and 1. The beta regression is then performed using this ODR. We would also like to clarify here that the *betareg* function in R that performs the beta regression uses a logit transformation of the values (mapping values from a (0, 1) interval to a $(-\infty, +\infty)$ interval) to perform the regression.

- 1.9 This first paragraph of Results directs us to look at Figure 1a, Figure 1b, and Figure 1c, meaning we are supposed to look at 27 panels, each with 6 layers. At the same time, we are meant to look at a supplementary figure and we are pointed to 8 more supplementary

figures. We are barely told what to look for in these figures. The result is completely overwhelming. Please limit the calls to figures to the minimum necessary in this paragraph, and let the next paragraph do its job (which it does well), namely to guide the reader in the comprehension of feedforward circuits (figure 1a) and thence to the rest.

We have revised the Results section and the supplementary figures to more clearly and appropriately cite and describe the corresponding figures. We now first focus on Supplementary Fig. 1, and describe in detail the injection methodology and the identification of areas based on anatomical landmarks (lines 102–110). We have revised Supplementary Fig. 1a to showcase an injection in AM because it more clearly shows the bisbenzimidazole-labelled callosal neurons. We have also combined the previous Supplementary Figures 2–9 into two figures: Supplementary Figures 2 and 3. We point out the diverse laminar patterns formed by the numerous interareal projections, and then describe the stereotypic properties of FF and FB projections involving V1 (lines 111–132).

- 1.10 Figure 1. This will sound like a drastic suggestion but hear it out: consider inverting the color of the anatomical images. This reviewer tried this inversion to check that it looks good, and it does look very good. The main advantage is that one does not have large black squares, which have high contrast relative to the white page but carry no information.

While we agree that a white background would be preferable (particularly for printing on white paper), and that many of the images did look good when the colors were inverted, we often noticed artifacts of color inversions in several images that made them inadequate for publication. We would also like to point out that the analysis considers pixel values (brightness) from each image, which is more intuitive if the pixels of labeled axons are white and those in the background are dark.

- 1.11 Figure 1. Next to each anatomical image, please provide the index computed for that specific image (which has to be something better than ODR, as explained above).

This is an excellent suggestion that illustrates the use of the ODR metric. We have now included the newly defined ODR in each panel of Fig. 1b–d. We wish to clarify here that the ODRs provided in Fig. 1b–d correspond to the individual image, not to the average of multiple patterns of the same pathway, and are therefore different than those in Fig. 2a.

- 1.12 “encompassed by L2-4”. In those images one sees activity throughout cortex with only exceptions being L1 and perhaps a hole in L4 (is it significant? It seems to be but this is not explained in the paper). So, it is not clear why this sentence focuses on L2-4 (as a reader, one is confused). Better to first point out the wide expression before L1 and then go in the details.

The reviewer has correctly noted that FF projections often show a distinct sparseness in lower L4 and upper L5, with the differential targeting of L1 by FF and FB pathways being the major contributor to the differences in ODR. We have accordingly revised the text to

indicate that the ODR was chosen because it showed a clear distinction between FF and FB pathways, that FB terminations typically showed a bistratified pattern with preferential targeting of L1 and L5/6, and that L5 and L6 were excluded from analysis because of axons of passage traversing through them (lines 129 to 142).

- 1.13 Figure 2a. It's clear why there are no values in the diagonal but: could there be values perhaps? It would be interesting to know what the values of the index are for intra-areal connections.

The reviewer raises an interesting point. We have excluded the ODR for intra-areal connections for primarily three reasons: (1) except for V1, LM, and PM, visual areas were too small to identify regions within them that were at a significant distance away from the injection site; (2) these values would not be used for the analyses of interareal connections, and would therefore be unsuitable for inclusion in the present study; and (3) identifying a termination zone for intra-areal projections is not trivial due to the presence of axons of passage that traverse to other areas after diving into L5, L6, and the white matter, and hence requires additional criteria and investigations that we consider to be beyond the scope of this study.

- 1.14 This expectation (“if a pathway is feedforward in one direction, the reciprocal...”) is actually confirmed by the data, and this is one of the strongest points of this paper. Consider emphasizing this in abstract, introduction, etcetera.

We thank the reviewer for this suggestion. We have now alluded to this finding in the Abstract (lines 27–28), and have more prominently described it in the Introduction (lines 78–81) and the Results (177–179, 293–297, 314–316; Figs. 2c and 4a, Supplementary Figs. 6c, d).

- 1.15 Figure 2. The scatter plots in Fig 2c are too small and numerous to follow. Please combined them into a scatter plot with all the data, combined, showing one dot per connection. Perhaps color the plots according to the areas of origin and of destination (for instance, the fill color could indicate the area of origin, and the outline color could indicate the area of destination). If this ends up being confusing, break it up into a few scatter plots. For instance, one scatter plot for the first column of 2c, one for the next column, and so on (that's 9 scatter plots instead of 45 scatter plots).

In the triangle plot (now Fig. 2d), each scatterplot shows that it is possible to identify a hierarchical distance measure between two injected areas based on the projection patterns to each of their common targets. The scatterplots do not identify every common target area of the two injected areas, but simply show that the y-intercept of the unit slope that best fits the data points acts as a distance measure between the two injected areas. This scatterplot organization is similar to ones that have been previously published for the primate brain in which a different anatomical index was used (e.g., Markov et al., 2014; Vezoli et al., 2021), and we intended to maintain the figure structure for easy comparisons across species. We have moved the explanation of these scatterplots to the Methods section under the subheading “*Pairs plot (Fig. 2d)*” (line 544). Additionally, we have now

included two representative scatter plots at higher magnification (Fig. 2d) to better explain the figure. We have also included another set of scatterplots in the new Supplementary Fig. 4, which instead determines hierarchical distance measures between two *target* areas based on termination patterns of pathways from all common source areas. These figures demonstrate a consistency of hierarchical rules across the network.

References:

Markov NT et al. (2014) Anatomy of hierarchy: feedforward and feedback pathways in macaque visual cortex. *J Comp Neurol.* 522(1):225-59. doi: 10.1002/cne.23458.

Vezoli J et al. (2021) Cortical hierarchy, dual counterstream architecture and the importance of top-down generative networks. *Neuroimage* 225:117479. doi: 10.1016/j.neuroimage.2020.117479

1.16 Figure 2c, rescale the axes to have equal ranges, so that the identity line is at 45 degrees. Lines 343-344.

We apologize for any confusion regarding Fig. 2c of the previous version of the manuscript. In the triangle plot (Fig. 2d of the new version), the axes have equal ranges, and the identity line in each panel is plotted so that it best fits the data points. The y-intercept of this identity line (on the dotted axis) gives us a measure of the hierarchical distance separating the two areas of each plot. In the scatterplot showing the relationship between all reciprocal connections (Fig. 2c of the new version of the manuscript), the solid black line is not an identity line, but is a linear regression line that best fits the data points. The axes are scaled so that all the data are included within the scatter plot. The statistically significant negative slope of this line indicates the negative association between reciprocal connections (i.e., the more FF a pathway is in one direction, the more FB it is in the opposite). We have explained these figures in more detail in the main text.

1.17 These two equations are unnecessary (by adding equations for things that are trivial, one risks losing readers).

We agree, and have revised the text so that the explanation of the triangle plot is now moved to the Methods section (lines 544–564), and have excluded the beta distribution description in the main text.

1.18 The “inverse logistic transformation of the natural log of the ODR” is a sentence that perfectly illustrates the strangeness of the approach, and so is the equation below it. Please just choose a reasonable index to begin with, as described above, and stick to it throughout the paper. It is also not clear at all that we need to hear about beta distributions. Where are the fits of the distribution to the data? What are we doing with the parameters of the fit? If they are not needed, please drop the whole concept. Just use a reasonable index and show that its distribution is bell-shaped.

We have fully revised the analysis so that the ODR is now defined as $L2-4/(L1 + L2-4)$ from the onset, and have excluded the descriptions of the beta regression.

- 1.19 Figure 3. By the time we have reached this figure we have seen ODR, $\log(\text{ODR})$, $\text{ODR}/(1+\text{ODR})$, and its (more understandable) synonym $L2-4/(L1-L2-4)$. The latter two appear in the same figure. This is not a good idea. Please choose a reasonable index from the outset, and stick to it.

While the new $L2-4/(L1 + L2-4)$ ODR replaces the old $L2-4/L1$ ODR, a logit transformation of this ratio is needed to maintain the linearity of hierarchical relationships between areas. This is now more clearly explained in the text (lines 161–165). We have also revised Fig. 2 and 3 so that the frequency distribution of logit transformed values of the ODRs are now shown in Fig. 2b and the estimated vs measured ODR plot is shown in Fig. 3b.

- 1.20 The Discussion (and to some extent also the Introduction) is verbose. Consider trimming it.

We have extensively revised the Introduction and Results sections.

- 1.21 The supplementary figures need headings and labels and could be streamlined and combined. For instance, one could put all the connections from one area to another in a column, and put multiple destination areas on the same page (e.g. 9 rows and 5 columns in one page, and 9 rows and the remaining 4 columns in the next page). This would not only reduce the number of supplementary figures but also make them easier to navigate.

We found this suggestion to be very helpful, and have accordingly reorganized the supplementary figures. Supplementary Figs. 2–9 of the original version of the manuscript have been combined into Supplementary Figs. 2 and 3. We have also included headings for the supplementary figures, and have appropriately labeled them.

Reviewer #2

- 2.1 The current study investigates this topic with smaller datasets using tracer injections and a small number of single unit recordings from anesthetized mice. Although the topic is important, I have the following major concerns. With its limited analyses and datasets, the manuscript is of interest to limited specialists, and not to the general readers of Nature Communications.

1. As one of the main conclusions of this study, the authors proposed an overlapping hierarchy levels (V1, LM, RL and the rest) with canonical and non-canonical network motifs. This proposal is of potential interest, but readers are left without any actual support. The study showed how the receptive field size changes in the hierarchical

organization, and how it can be captured by the ventral and dorsal streams. This can be consistent with canonical motifs. However, the relation between this size change and the four hierarchy levels proposed in this study is not clear. Nor is clear its relation with the lateral connections. The proposal on the non-canonical network remains as a hypothesis that cannot be tested.

Our anatomical results indicate that the mouse visual network comprises a large number of “lateral” connections, and therefore should not be considered a purely hierarchical sequence of areas. The Akaike information criterion (AIC) was used to determine the number of levels that provides the best predictive power for the inclusion of new data (e.g., projections to and from an additional area within the network). We have now modified the old Fig. 4c (Fig. 3g of the new manuscript) to directly compare the hierarchy based on an anatomical metric with differences in RF diameters. We find that while comparisons between RF sizes of different areas are not perfectly predictive of their relative hierarchical positions (Fig. 3g), overall the pairwise comparisons of RF diameters and the increase in RF diameters along each processing stream show a remarkable consistency with the anatomical hierarchy (Fig. 3a vs 3f; Fig. 3g). Additionally, the functional consequences of canonical and noncanonical circuits are likely to be manifested not only in RF dimensions but also in other “higher-order” factors as well such as contextual modulation (e.g., through surround suppression) and predictive signals, which as the reviewer correctly points out, cannot be easily tested. Our results however provide a significant advance to the notion of a hierarchical organization in the mouse brain by differentiating canonical and noncanonical circuits.

Please see also our response to reviewer #1 in section 1.2.

2.2 2. Their physiological measurements address an important question, how visual representation is transformed along the hierarchical ladder. However, there are three concerns. First, it is not clear why the findings here are not consistent with the report from the same lab (Wang and Burkhalter 2007, JCN). In their previous report, the order of the receptive field sizes were V1<LM<POR<AL<P<LI<PM<AM<RL<A. How do the study reconcile these differences? Second, there are many other receptive field parameters to compare, for instance, latency, linearity, etc. Those parameters should be reported in their relationship with the hierarchical organization. Third, the data sample number is too small ($n < 10$) in each area (except for V1 and LM). $N = 10$ cannot be sufficient, considering the possibility of layer and cell-type differences in a given area. Therefore, although the physiological data are interesting preliminary findings, I would expect more complete datasets and analyses for publication.

The reviewer raises an important point with regard to the differences in RF sizes in the present study and those in Wang and Burkhalter, 2007. In the 2007 study, RF mapping was performed through a qualitative assessment of spiking by listening to an audiometer, and RF positions and sizes were mapped manually onto a spherical dome used as the projection screen. In the present study RF sizes were measured quantitatively. In addition, RFs in the 2007 study were plotted using high contrast bars and edges as

stimuli, which differed from the drifting gratings used in the present study. It seems likely that the discrepancies between the two studies are due to methodological differences.

We agree that additional RF parameters would have strengthened the study and complemented the anatomical hierarchy. We regrettably did not record these parameters; however, a recently published study did perform similar recordings albeit in not all of the areas examined in the present study (Siegle et al., 2021). While RFs recorded in our study were smaller than those in Siegle et al., both studies show an increase in RF sizes with increasing hierarchical levels. Please also see our responses to reviewer #1 in section 1.2.

We would also like to clarify that our single-unit recordings were performed in L2/3, which have been shown to have smaller RFs than those in deep layers (e.g., Self et al., *J Neurosci* 2014), whereas Siegle et al. pooled RFs from all layers. RFs in our study were also optimized for high-contrast gratings, which may have contributed to the smaller RF sizes. These RF sizes however are comparable to other previous studies in both anesthetized and awake animals (Self et al., *J Neurosci* 2014; Keller et al., *Nature* 2020; Van den Bergh et al., 2010). The discrepancy in RF sizes could be attributed to a number of factors including the layers from which the cells were sampled, anesthesia, and visual stimulus properties such as contrast and size. Notably, Siegle et al. (2021) used a 20 deg diameter drifting grating stimulus making it unlikely for them to be able to measure RFs smaller than 20 deg.

We have increased the number of RF mapping recordings, and taking into account reviewer #1's comments, we have minimized the role of the physiological data and combined them with the hierarchical organization in Fig. 3.

References:

Self MW et al. (2014) Orientation-tuned surround suppression in mouse visual cortex. *J Neurosci* 34(28):9290-304. doi: 10.1523/JNEUROSCI.5051-13.2014

Keller AJ et al. (2020) Feedback generates a second receptive field in neurons of the visual cortex. *Nature* 582(7813):545-549. doi: 10.1038/s41586-020-2319-4

Van den Bergh G (2010) Receptive-field properties of V1 and V2 neurons in mice and macaque monkeys. *J Neurosci* 30(11):2051-70. doi: 10.1002/cne.22321

2.3 1. Lateral connections.

Lateral connections are the key to this study. The connections, however, are basically an unclassified group, that cannot be identified as either feedforward or feedback. As mentioned in the discussion, the connections can include reciprocally connected pairs of areas, exhibiting feedforward and feedback termination patterns in both directions. The study should prepare a figure or supplemental figure that shows various types of termination patterns in both directions.

We agree with the reviewer that the widespread presence of lateral connections is a central finding of this study. We have now briefly described various patterns of interareal connections in the Results section (lines 111–122), whereas Fig. 1b–d, and Supplementary Figs. 2 and 3 demonstrate the diversity of these patterns. We would like to emphasize, however, that the approach used in the present study to build the hierarchy is fundamentally different than one in which individual projection patterns are classified into FF, FB, and lateral categories (e.g., Felleman and Van Essen, 1991; Harris et al., 2019). Our approach to building the hierarchy was based not on pairwise comparisons of laminar patterns between reciprocally connected areas, but on connections to all target areas within the network. As such, the diverse types of laminar patterns across pathways were not used to construct the hierarchy, but rather, whether a pathway is FF or FB or lateral was determined based on the hierarchical positioning of the corresponding areas (which was determined by the ODR). We are therefore of the opinion that including a figure categorizing the various types of termination patterns would be incongruent with the analysis performed to build the hierarchy.

2.4 2. Difference between this study and Allen Institute's report in 2019

Allen Institute uses different categorization of feedforward and feedback projections for laminar distributions. In addition, Allen's report included thalamo-cortical, and cortico-thalamic projections to define their hierarchy. Please clarify the advantage and disadvantage of the current study over the one by Allen Institute.

We have now emphasized the differences between the present study and Harris et al. (2019) in the Introduction and Discussion sections (lines 59–67, 68–76, 325–334, 341–348).

2.5 3. Densest 70% of projections

The conclusions of the study is built on the measurement of average optical density of projections. The study uses the densest 70% of projections. It is important to show whether the conclusions remains the same with different criterion (e.g., 90%, 50%).

We found this to be a highly valuable suggestion for showing the robustness of the hierarchy. We therefore reanalyzed all of the anatomy data by using a custom-written script for measuring the ODR after selection of the top n% brightest pixels (i.e. pixels with the highest n% optical densities) from each dark-field image. While the majority of the findings were based on the selection of the top 70% pixels, we have also included the results of the hierarchical analyses after inclusion of the top 10% and top 90% optical densities. Notably, the hierarchy was invariant across the optical density thresholds chosen.

2.6 4. Please consider a schema of the injection site and target sites for figures.

It will be helpful for the general readers to prepare a schema for each anatomical experiment as supplementary figures.

We agree, and have included a schematic diagram in Fig. 1a and Supplementary Fig. 1b.

REVIEWER COMMENTS

Reviewer #1 (Remarks to the Author):

I remain enthusiastic about this paper. The rebuttal did not include the "positive" part of my review, and it may be worth pasting it here for all to see: "This wonderful and important study shows that there is a clear logic to the connections between visual areas in the mouse cortex. The background of the study is that the laminar pattern of axons sent from one area to another displays typical features if it is a feedforward connection (axons avoid layer 1) or a feedback connection (axons favor layer 1). Based on this fact, the study reveals a remarkable finding: that the more "feedforward-looking" the connection in one direction, the more "feedback-looking" the connection in the other direction. Moreover, this internal logic to the connections allows the authors to build a hierarchy of visual cortical areas in the mouse that is in many ways as compelling as similar hierarchies that have been drawn for primates."

The revision has substantially improved. For instance, it no longer uses 3-4 different indices to measure the same thing, etc. Below are some suggestions for further improvements, which are just suggestions – they should not be considered to be prescriptive.

Content

- One of the key messages of the paper is that the areas in the mouse visual cortex are arranged lawfully: the more "feedforward-looking" the connection from area A to area B, the more "feedback-looking" the connection from B to A. This is a fundamental finding and it should be given more emphasis in abstract, in introduction (emphasizing the excellent text in lines 72-74 and line 77), in results (emphasizing the language in lines 178-179, and the excellent analysis in lines 182 and subsequent), and in Discussion.
- Another key message that is not coming out in abstract is that the ventral pathway appears to be much more hierarchical than the parietal pathway (results in lines 296-297)
- Lines 37-39: this is a somewhat esoteric list of processes, and it omits the basics of what the visual system does in the first 100 ms of vision: image processing via filtering (e.g. DiCarlo, J.J., Zoccolan, D., and Rust, N.C. (2012). How does the brain solve visual object recognition? *Neuron* 73, 415-434; Yamins, D.L., and DiCarlo, J.J. (2016). Using goal-driven deep learning models to understand sensory cortex. *Nature Neuroscience* 19, 356-365).
- The first 3 paragraphs of Results ask that we look at Supplementary Figure 1. It is thus an important figure, and indeed it is a lovely figure. Why not promote it to main text? It would greatly help the paper.
- Line 117 then asks that we look at 2 more supplementary figures. We have not even looked at figure 1, and we are supposed to have digested 3 supplementary figures? This is not particularly helpful to a reader. Usually the main text figures make the points that the paper wants to make, and the supplementary figures add some extra information that particularly interested readers may want to look at. It is not a good idea to use them to make key points, such as "strikingly different laminar differences across pathways".
- Fig 2c. It's hard to relate the logit values to the original ODRs. Consider showing both as tick marks (on two different axes).
- Lines 391-294. This sentence seems to contradict the paper! The paper relies on (relative) projections to L1 to measure the feedback nature of a connection. This sentence says that projections to L1 are characteristic of all kinds of projection? This is very odd. Please edit this sentence or simply drop it.
- Lines 399-411 This entire paragraph is speculation. It does not progress the paper in any way. Please consider dropping it. Indeed, the paper says nothing about predictive coding, and predictive coding says little or nothing about the arrangement of areas.

Style

- Line 23: remove "active". Besides being unnecessary, it is misleading given that this paper is about brains that are sectioned or anesthetized (information that does not appear in the abstract).
- The first 3 sentences of the abstract don't connect to each other.
- The reference to "canonical" and "noncanonical" network motifs in abstract is going to be obscure to 99% of readers. The definitions are given in lines 52 and 54, from which one evinces that "canonical" (9 letters) means "hierarchical" (13 letters), and "noncanonical" (13 letters) means

“not hierarchical” (17 letters). So to save 8 letters we need to use two vague words that most readers will not know? The title of course could change too.

- Lines 273 and following: if the error in RF diameter (presumably the standard error of the mean? Please clarify) is >1 degree, we don't need a precision of two decimals in specifying these diameters.

- “feedforwardness” and “feedbackness” are not English words. Consider replacing with “feedforward or feedback nature”.

Reviewer #2 (Remarks to the Author):

The manuscript has improved, and some of the issues were clarified. However, there remain several critical concerns. Most importantly, the claim of the existence of “noncanonical” network is merely based on a statistical analysis (AIC) using ODR. To make the claim more convincing, the study needs to show additional independent evidence to demonstrate that non-canonical and canonical networks are indeed distinct from each other.

2.1 Definition of “non-canonical”

A previous study from Allen Institute (Harris et al., 2019) has sorted visual areas based on the hierarchical levels, and demonstrated that some of visual areas have a similar Hierarchy score. This study reports a conceptually similar finding - some areas have similar Hierarchy score (e.g., P, AL, A and PM in Figure 3d). The novelty of the paper is the claim that the connections between such areas are distinct from the “canonical network”. However, the supporting evidence for this claim is limited to AIC analysis. To demonstrate noncanonical network are distinct type of long-range connections, it is necessary to show an independent (physiological or histological) evidence, beyond just relying on the outcome of the statistical test. Conclusions based on a single analysis might be problematic; they can be potentially important but without more convincing evidence the manuscript is more suitable for specialized journals rather than a journal for general readers.

2.3, 2.4. Axon terminal patterns and the relationship with Harris et al, 2019.

Allen Institute's paper (Harris et al (2019)) has classified axon terminal patterns across all the layers into 9 groups using unsupervised hierarchical clustering (see their Fig 5). Their classification is based on much larger number of data-set than this manuscript.

This study, in contrast, condensed 6 dimension vector (labeling in each of the 6 layers) into a single value $ODR = L_{2,3,4} / (L_1 + L_{2,3,4})$. It is not clear how such condensed value can be justified when a more thorough work reported 9 different patterns of axon terminals. Note that Harris et al. (2019) classified cluster I that projects predominantly to layer I (therefore low ODR) as feedforward. As the entire analysis of this paper depends on the premise that ODR is an accurate indicator of FF and FB connections, a more clear justification of this variable is expected.

2.2. Inconsistency with the author's past study (Wang and Burkhalter, 2007).

We raised this issue in the previous review. The manuscript needs to have a discussion on the discrepancy with their own past publication (Wang and Burkhalter 2007). In their previous report, the order of the receptive field sizes were $V1 < LM < POR < AL < P < LI < PM < AM < RL < A$. It is unprofessional not to mention the discrepancy with the publication from the same lab.

2.2. RF properties.

It is not clear why the authors do not want to study more details of the RF structures, which could potentially help them identify non-canonical network at the functional level. Limited effort in physiological recordings reduce the general interest of the manuscript.

Point-by-point responses to the reviewers' comments

Reviewer #1

I remain enthusiastic about this paper. The rebuttal did not include the “positive” part of my review, and it may be worth pasting it here for all to see: “This wonderful and important study shows that there is a clear logic to the connections between visual areas in the mouse cortex. The background of the study is that the laminar pattern of axons sent from one area to another displays typical features if it is a feedforward connection (axons avoid layer 1) or a feedback connection (axons favor layer 1). Based on this fact, the study reveals a remarkable finding: that the more “feedforward-looking” the connection in one direction, the more “feedback-looking” the connection in the other direction. Moreover, this internal logic to the connections allows the authors to build a hierarchy of visual cortical areas in the mouse that is in many ways as compelling as similar hierarchies that have been drawn for primates.”

The revision has substantially improved. For instance, it no longer uses 3-4 different indices to measure the same thing, etc. Below are some suggestions for further improvements, which are just suggestions – they should not be considered to be prescriptive.

We thank the reviewer for their positive comments on the manuscript. Where applicable, we have revised the manuscript in accordance with the reviewer's suggestions, as described below:

- One of the key messages of the paper is that the areas in the mouse visual cortex are arranged lawfully: the more “feedforward-looking” the connection from area A to area B, the more “feedback-looking” the connection from B to A. This is a fundamental finding and it should be given more emphasis in abstract, in introduction (emphasizing the excellent text in lines 72-74 and line 77), in results (emphasizing the language in lines 178-179, and the excellent analysis in lines 182 and subsequent), and in Discussion.

We have now emphasized this finding in the Introduction (lines 76-78), Results (lines 176-178), and Discussion (lines 353-354). We believe the line “Reciprocally connected pairs of areas exhibited feedforward/feedback relationships consistent with a hierarchical organization” in the Abstract alludes to this finding. However, we also want to highlight the fact that despite this negative correlation between FF and FB pathways, the network is characterized by numerous lateral connections that are inconsistent with a strict hierarchy comprised exclusively of FF and FB pathways.

- Another key message that is not coming out in abstract is that the ventral pathway appears to be much more hierarchical than the parietal pathway (results in lines 296-297)

We have described this finding in the Abstract (lines 32-33) and Results (lines 279-280).

- Lines 37-39: this is a somewhat esoteric list of processes, and it omits the basics of what the visual system does in the first 100 ms of vision: image processing via filtering (e.g. DiCarlo, J.J.,

Zoccolan, D., and Rust, N.C. (2012). How does the brain solve visual object recognition? *Neuron* 73, 415-434; Yamins, D.L., and DiCarlo, J.J. (2016). Using goal-driven deep learning models to understand sensory cortex. *Nature Neuroscience* 19, 356-365).

We have revised the first sentence to include object recognition, and have cited DiCarlo et al., 2012.

- The first 3 paragraphs of Results ask that we look at Supplementary Figure 1. It is thus an important figure, and indeed it is a lovely figure. Why not promote it to main text? It would greatly help the paper.

- Line 117 then asks that we look at 2 more supplementary figures. We have not even looked at figure 1, and we are supposed to have digested 3 supplementary figures? This is not particularly helpful to a reader. Usually the main text figures make the points that the paper wants to make, and the supplementary figures add some extra information that particularly interested readers may want to look at. It is not a good idea to use them to make key points, such as “strikingly different laminar differences across pathways”.

We agree, and the previous Supplementary Fig. 1 is now presented as Fig. 1 in the main text. Thus, Figs. 1 and 2 of the revised manuscript provide essential information regarding the differences in laminar patterns, and Supplementary Figs. 1 and 2 provide additional examples.

- Fig 2c. It's hard to relate the logit values to the original ODRs. Consider showing both as tick marks (on two different axes).

Supplementary Fig. 3a of the revised manuscript plots the logit transformation of the ODR against the ODR. Supplementary Fig. 3b is similar to the scatter plot of Fig. 3c (Fig. 2c of the previous submission), except it plots the raw ODRs instead of the logit values.

- Lines 391-294. This sentence seems to contradict the paper! The paper relies on (relative) projections to L1 to measure the feedback nature of a connection. This sentence says that projections to L1 are characteristic of all kinds of projection? This is very odd. Please edit this sentence or simply drop it.

We intend to highlight the finding that L1, which is often believed to be the main target of FB projections, is actually a common target of virtually all pathways in higher visual cortex. While the targeting of L1 relative to that of L2-4 provides a hierarchical measure, the absolute strength of connections to L1 does not. We believe this is an important point for the study to explicitly make, and have revised the corresponding lines (426-430).

- Lines 399-411 This entire paragraph is speculation. It does not progress the paper in any way. Please consider dropping it. Indeed, the paper says nothing about predictive coding, and predictive coding says little or nothing about the arrangement of areas.

We have deleted this paragraph from the manuscript.

Style

- Line 23: remove “active”. Besides being unnecessary, it is misleading given that this paper is about brains that are sectioned or anesthetized (information that does not appear in the abstract).

We have deleted “active”, and now indicate that recordings were done in anesthetized mice (line 31).

- The first 3 sentences of the abstract don't connect to each other.

We have revised the abstract to improve its flow.

- The reference to “canonical” and “noncanonical” network motifs in abstract is going to be obscure to 99% of readers. The definitions are given in lines 52 and 54, from which one evinces that “canonical” (9 letters) means “hierarchical” (13 letters), and “noncanonical” (13 letters) means “not hierarchical” (17 letters). So to save 8 letters we need to use two vague words that most readers will not know? The title of course could change too.

We have revised the manuscript, including the title, to replace ‘canonical’ with ‘hierarchical’ and ‘noncanonical’ with ‘nonhierarchical’.

- Lines 273 and following: if the error in RF diameter (presumably the standard error of the mean? Please clarify) is >1 degree, we don't need a precision of two decimals in specifying these diameters.

We have limited the RF diameter values to 1 decimal. We have also clarified in the Methods section that the diameters are presented as means \pm SEM (lines 540-541).

- “feedforwardness” and “feedbackness” are not English words. Consider replacing with “feedforward or feedback nature”.

We have made these changes throughout the manuscript.

Reviewer #2 (Remarks to the Author):

The manuscript has improved, and some of the issues were clarified. However, there remain several critical concerns. Most importantly, the claim of the existence of “noncanonical” network is merely based on a statistical analysis (AIC) using ODR. To make the claim more convincing, the study needs to show additional independent evidence to demonstrate that non-canonical and canonical networks are indeed distinct from each other.

2.1 Definition of “non-canonical”

A previous study from Allen Institute (Harris et al., 2019) has sorted visual areas based on the hierarchical levels, and demonstrated that some of visual areas have a similar Hierarchy score. This study reports a conceptually similar finding - some areas have similar Hierarchy score (e.g., P, AL, A and PM in Figure 3d). The novelty of the paper is the claim that the connections between such areas are distinct from the “canonical network”. However, the supporting evidence for this claim is limited to AIC analysis. To demonstrate noncanonical network are distinct type of long-range connections, it is necessary to show an independent (physiological or histological) evidence, beyond just relying on the outcome of the statistical test. Conclusions based on a single analysis might be problematic; they can be potentially important but without more convincing evidence the manuscript is more suitable for specialized journals rather than a journal for general readers.

In response to reviewer #1’s comments, we have revised the manuscript to refer to canonical and noncanonical circuits as simply hierarchical and nonhierarchical, respectively.

We would like to emphasize that the novelty of the present study includes the identification of an anatomical (histological) metric, the ODR, for quantifying the hierarchical nature of interareal connections. We have added Fig. 4f and 4g, which shows that the ODRs for FF, lateral, and FB connections are significantly different. We believe that our pairwise comparisons of RF diameters addresses the reviewer’s comments regarding lateral (noncanonical) connections: in the majority of cases, the relative hierarchical ordering of areas is consistent with differences in physiological properties, with areas at different hierarchical levels typically showing differences in RF diameters, and those at the same level showing dissimilar RF sizes.

We have now expanded this finding to open-source data obtained from the Allen Institute, recorded from awake, head-fixed mice. The reviewer is correct that the hierarchy score of the Allen Institute is conceptually similar to the hierarchical levels of the present study. We show this more directly in the new Supplementary Fig. 5a, which shows a strong correlation between the hierarchy score of Harris et al., 2019 and our hierarchical levels. This is an important finding because it implies a robustness of the ODR in constructing cortical hierarchies that is consistent with one that was built using a clustering analysis of thousands of laminar patterns (Harris et al., 2019).

We further show that the hierarchical levels of the present study are consistent with the physiological recordings (RF size and spike latency) of Siegle et al., 2021 (Supplementary Fig. 5b, c).

We have also revised the goodness-of-fit plots in Fig. 4b and Fig. 5d to highlight the point that the logit of the ODR acts as a measure of hierarchical distance. This implies that the laminar termination patterns from a single anterograde injection is predictive of the injected area’s position in the cortical hierarchy. This is a novel finding, and we believe it be a significant advance from Harris et al., 2019.

2.3, 2.4. Axon terminal patterns and the relationship with Harris et al, 2019.

Allen Institute's paper (Harris et al (2019)) has classified axon terminal patterns across all the layers into 9 groups using unsupervised hierarchical clustering (see their Fig 5). Their classification is based on much larger number of data-set than this manuscript.

This study, in contrast, condensed 6 dimension vector (labeling in each of the 6 layers) into a single value $ODR = L_{2,3,4} / (L_1 + L_{2,3,4})$. It is not clear how such condensed value can be justified when a more thorough work reported 9 different patterns of axon terminals. Note that Harris et al. (2019) classified cluster I that projects predominantly to layer I (therefore low ODR) as feedforward. As the entire analysis of this paper depends on the premise that ODR is an accurate indicator of FF and FB connections, a more clear justification of this variable is expected.

We have added Supplementary Fig. 5a, which shows a strong correlation between the Allen Institute's hierarchy score and the hierarchical level of the present study. This shows that the ODR, a single parameter that only considers terminations in layers 1 to 4, is a robust index for quantifying the hierarchical nature of interareal pathways.

Furthermore, we have also measured the mean ODRs for FF, lateral, and FB pathways (Fig. 4f, g), with FF pathways having a mean ODR of 0.57 ± 0.02 . This does indicate stronger input to L2-4 than to L1 for FF pathways. Notably, the strategy used by Harris et al., 2019 was to minimize (but which did not eliminate) the number of violations of the rule that a FF connection is reciprocated by a FB connection. This led to multiple violations of the hierarchical rule, leading to a low 'global hierarchy score' (Fig. 6c of Harris et al., 2019). Arguably, a reason for so many violations was the exclusion of the notion of lateral connections from the analysis. For example, the FB AL→LM and A→LM pathways of the present study (Supplementary Fig. 2) resemble cluster 1 of Harris et al., 2019, where they might be considered FF, even though AL (VISal) and A (VISa) have a higher hierarchy score than LM (VISl) in the same study. The clusters also appear to lack a pattern in which all six layers or layers 1-5 were targeted with relatively equal strengths, even though this was a common observation in our study (e.g., POR→P, P→LM, P→AL, LM→AL). Thus, while the clustering analysis of Harris et al., 2019 provides valuable insight into the hierarchical characteristic of the mouse visual network, its investigation into lateral connections is lacking, and it lacks the provision of a hierarchical index that scales with the hierarchical nature of connections that the present study provides.

2.2. Inconsistency with the author's past study (Wang and Burkhalter, 2007).

We raised this issue in the previous review. The manuscript needs to have a discussion on the discrepancy with their own past publication (Wang and Burkhalter 2007). In their previous report, the order of the receptive field sizes were $V1 < LM < POR < AL < P < LI < PM < AM < RL < A$. It is unprofessional not to mention the discrepancy with the publication from the same lab.

We apologize for this oversight, and have included an explanation on why the ordering of RF sizes differs between the two studies (lines 385-391)

2.2. RF properties.

It is not clear why the authors do not want to study more details of the RF structures, which could potentially help them identify non-canonical network at the functional level. Limited effort in physiological recordings reduce the general interest of the manuscript.

We regrettably did not record physiological properties other than RF diameters while acquiring our single-unit data. We therefore obtained RF size and spike latency data from the study by Siegle et al., 2021, and compared it with our hierarchy. The results show a high level of consistency between the two hierarchies (Supplementary Fig. 5b, c).

REVIEWER COMMENTS

Reviewer #2 (Remarks to the Author):

The manuscript now reads smoothly. At the same time, it has become clearer that the novelty of the manuscript is limited, given that Harris et al., 2018 has reported conceptually similar findings. While there are some new aspects in the findings/analysis of the current manuscript, the paper is more suited for publication in a specialized journal rather than in Nature Communications, where we expect a paper of general interest.

1) "that the novelty of the present study includes the identification of an anatomical (histological) metric, the ODR"

Although ODR is an attractive metric, it is conceptually similar to the metric that was already published in the authors' previous paper 8 years ago (Markov et al., 2013) (with the difference of using anterograde vs. retrograde tracers). The analysis procedures are also almost identical to the ones published in the same paper, and thus, the analysis is not novel. Ultimately, the metric for hierarchy reported in this study (hierarchical level) is very similar to the "hierarchical score" in Harris et al., 2018 (as the authors acknowledged in the manuscript), and therefore does not seem to advance our understanding of the mouse visual system in a way expected from a publication in Nature Communications.

2) Definition of non-hierarchical.

The authors have added new analyses on non-hierarchical circuits to demonstrate internal consistency of their procedures. However, it remains unclear whether there is a qualitative difference between hierarchical and non-hierarchical circuits. Could it be the case that the non-hierarchical circuits reflect the two areas where the differences in the ODR are merely smaller than the noise level of the experiments? In any case, I would expect a more careful estimation on the reliability of the two critical values in the paper - the ODR and the hierarchical level.

2-1) I suggest the authors split the data into two groups, and test whether the same hierarchical patterns will be observed in each group. This will show the reliability of their analysis.

2-2) Could the authors estimate the variability of ODR across different sections (antero-posterior levels) and different mediolateral levels within the sections? Could the sections at the center of the projection have different ODR compared to those at the edge of the projection?

3) Receptive field size

a) The manuscript states "The hierarchical ordering of RF sizes in the present study also differs slightly from that in our previous report."

However, I would not call the difference "slight." Some of the key orders of the RF size are reversed ($V1 < LM < POR < AL < P < LI < PM < AM < RL < A$ in their previous paper). Could the authors plot the relationship between the hierarchical level and the RF diameter based on their previous paper? Please add the results of this analysis as a subfigure in Fig 4j and Fig 5S. Is the correlation significant?

b) The receptive field size may correlate with the hierarchical level, but it does not tell us whether the connections are hierarchical or non-hierarchical. For example, in the dorsal pathway, the RF size difference between AL and PM (same hierarchical level) are larger than the RF size difference between A and RL (different hierarchical level) (Fig 4d, j). Do the authors have any physiological support for the existence of non-hierarchical connections?

Minor points:

The details of the methods are not clear. For example, line 571: These background intensity values were respectively subtracted from all pixel values in L1 and L2-4.

The smallest values for L1 and L2-4 look different in Fig 2a and Fig 6S. Are the authors using different backgrounds for L1 and L2-4? It would be more natural to use the same normalization procedures for L1 and L2-4.

Could the authors also describe how they confirm the pixels are not saturated?

Reviewer #2 (Remarks to the Author):

The manuscript now reads smoothly. At the same time, it has become clearer that the novelty of the manuscript is limited, given that Harris et al., 2018 has reported conceptually similar findings. While there are some new aspects in the findings/analysis of the current manuscript, the paper is more suited for publication in a specialized journal rather than in Nature Communications, where we expect a paper of general interest.

1) “that the novelty of the present study includes the identification of an anatomical (histological) metric, the ODR”

Although ODR is an attractive metric, it is conceptually similar to the metric that was already published in the authors’ previous paper 8 years ago (Markov et al., 2013) (with the difference of using anterograde vs. retrograde tracers). The analysis procedures are also almost identical to the ones published in the same paper, and thus, the analysis is not novel. Ultimately, the metric for hierarchy reported in this study (hierarchical level) is very similar to the “hierarchical score” in Harris et al., 2018 (as the authors acknowledged in the manuscript), and therefore does not seem to advance our understanding of the mouse visual system in a way expected from a publication in Nature Communications.

We have identified for the first time a gradual anatomical metric for evaluating hierarchical distances in the mouse, a species that is fundamental to cortical neuroscience research. Importantly, unlike the ‘hierarchical score’ (Harris et al., 2018), which was determined post-hoc, the ODR makes no assumption of a categorical FF/FB nature of individual pathways, and is predictive of hierarchical positioning of an area; for example, evaluation of laminar patterns in an eleventh cortical area not included in the present study would provide insight into its hierarchical position based on the ODRs, and would predict the laminar termination patterns of axonal projections from that area. The inclusion of lateral connections in the present study is, in our opinion, a significant advance to the hierarchical diagram of Harris et al., 2018, and suggests that a strict hierarchy may not be necessary for predictive processing (Keller and Mrsic-Flogel, 2018). Furthermore, the ODR metric emphasizes the importance of layer 1, a locus that plays a critical role in learning and memory (Shi et al., 2021).

2) Definition of non-hierarchical.

The authors have added new analyses on non-hierarchical circuits to demonstrate internal consistency of their procedures. However, it remains unclear whether there is a qualitative difference between hierarchical and non-hierarchical circuits. Could it be the case that the non-hierarchical circuits reflect the two areas where the differences in the ODR are merely smaller than the noise level of the experiments? In any case, I would expect a more careful estimation on the reliability of the two critical values in the paper - the ODR and the hierarchical level.

2-1) I suggest the authors split the data into two groups, and test whether the same hierarchical patterns will be observed in each group. This will show the reliability of their analysis.

We would like to clarify that our description of hierarchical relationships is that of a gradual change in the hierarchical nature of pathways from the most feedforward to the most feedback. Our analysis provides a quantification of this gradient nature of the network hierarchy. Nevertheless, if we do divide the pathways into FF/FB and lateral categories based on the hierarchical diagram (Figs. 4d, e), we are able to identify distinct physiological properties (Fig. 4j, Supp. Fig. 5) as well as consistencies with individual pathways (Figs. 4f, g).

With regard to splitting the data into two groups, we have shown in Fig. 5 that including only a subset of connections maintains a negative correlation between reciprocal connections, and we have presented the hierarchy of this subnetwork. Splitting the data into two relatively equal groups substantially decreases the number of connections in each network, flattening the hierarchy. We do not believe this addresses the reliability of the analysis.

2-2) Could the authors estimate the variability of ODR across different sections (antero-posterior levels) and different mediolateral levels within the sections? Could the sections at the center of the projection have different ODR compared to those at the edge of the projection?

Our analysis was performed from 3-5 coronal sections for each projection in a target area; thus, the standard error shown in Fig. 3a is a result of variability in the antero-posterior direction. To address the issue of whether the center and the edges of the projection have different ODRs, we performed new regression analysis after exclusion of the top 10% brightest pixels. This would eliminate the center of the projection. The results are shown in Supplemental Figs. 6i and 6j. The analysis shows only modest changes to both, the slope of the negative correlation between reciprocal connections and the hierarchical arrangement of areas, when the densest projections are eliminated.

3) Receptive field size

a) The manuscript states “The hierarchical ordering of RF sizes in the present study also differs slightly from that in our previous report.”

However, I would not call the difference “slight.” Some of the key orders of the RF size are reversed (V1<LM<POR<AL<P<LI<PM<AM<RL<A in their previous paper). Could the authors plot the relationship between the hierarchical level and the RF diameter based on their previous paper? Please add the results of this analysis as a subfigure in Fig 4j and Fig 5S. Is the correlation significant?

In Wang and Burkhalter, 2007, the RF sizes were measured qualitatively and recorded manually on a spherical (dome) screen by listening to an audiomonitor, as opposed to the quantitative, automated method used in the present study. At the time, the identification of areal borders was not standardized as it is today, with immunostaining of the M2 acetylcholine receptor not being employed for areal identification. Areal borders in the 2007 study were identified by a reversal in the topography of visual responses upon advancing the location of the recording electrode. It is important to note that map reversals are gradual transitions. Many of the recordings in the 2007 mapping paper were derived from such locations, and likely contributed to inaccuracies in areal identification and measurements of RF size. For example, RF size of POR neurons may have been underestimated because some recordings were in fact from LM whose RFs are smaller. Similarly, the mean RF size of RL may have been overestimated because of contamination by larger RFs from A. In addition, it is conceivable that RF size was influenced by the stimulus used for mapping. In the 2007 paper, a wide range of hand-held stimuli were used, anything that evoked a spiking response. This approach was blind to specific stimulus requirements, such as surround suppression or surround enhancement, and may have contributed to differences between studies.

We therefore believe that including the 2007 data in the present manuscript would be misleading and confusing to the reader. We agree, however, that the difference in the RF ordering is not minor, and have deleted the word “slight” from the relevant sentence (line 391).

b) The receptive field size may correlate with the hierarchical level, but it does not tell us whether the connections are hierarchical or non-hierarchical. For example, in the dorsal pathway, the RF size difference between AL and PM (same hierarchical level) are larger than the RF size difference between A and RL (different hierarchical level) (Fig 4d, j). Do the authors have any physiological support for the existence of non-hierarchical connections?

Evaluation of differences in the physiological properties of lateral connections vs. FF/FB connections would require recording the activity of neurons in each area upon activation of individual interareal pathways (e.g., by optogenetic stimulation of specific pathways). Using Channelrhodopsin2-assisted mapping, we have previously shown that the relative excitation of parvalbumin-positive (Pv+) interneurons and pyramidal cells by interareal pathways depends on the hierarchical positions of the source and target areas: the relative activation of Pv+ interneurons scales with the hierarchical nature of the interareal pathway from the most FF to the most FB (Yang et al., *J Neurosci*, 2013; D’Souza et al., *eLife*, 2016). We therefore expect that the relative excitation of Pv+ interneurons (and therefore the inhibition/excitation ratio) is at intermediate levels within lateral pathways compared to FF (highest) and FB (lowest) pathways.

Minor points:

The details of the methods are not clear. For example,

line 571: These background intensity values were respectively subtracted from all pixel values in L1 and L2-4.

The smallest values for L1 and L2-4 look different in Fig 2a and Fig 6S. Are the authors using different backgrounds for L1 and L2-4? It would be more natural to use the same normalization procedures for L1 and L2-4.

Could the authors also describe how they confirm the pixels are not saturated?

Yes, we used different backgrounds for L1 and L2-4. This was done to compensate for edge artifacts that often lead to a brighter background in L1 because of light incident onto the pial surface. It is expected that the smallest values in L1 and L2-4 differ due to the selection of the brightest n% of pixel values and the differences in the density of anterogradely labeled fibers in the two regions.

The exposure time for imaging was determined through an iterative process so that individual fibers could be identified without saturation of the image. The inclusion of the results of the analysis after exclusion of the top 10% pixels (Supplemental Figs. 6i and 6j) show that excluding the brightest pixels do not substantially alter the hierarchical organization.